

# Aqueous phase oxidation of bisulfite influenced by nitrate photolysis

Lu Chen[a], Lingdong Kong[a, b,*], Songying Tong[a], Kejing Yang[a], Shengyan Jin[a], Chao Wang[a], Lin Wang[a]

[a] Department of Environmental Science & Engineering, Jiangwan Campus, Fudan University, No. 2205 Songhu Road, Shanghai, 200438, China

[b] Institute of Eco-Chongming, East China Normal University, No.3663 Northern Zhongshan Road, Shanghai 200062, China

*Correspondence to:* Lingdong Kong (ldkong@fudan.edu.cn)


**ABSTRACT:** Nitrate aerosol is ubiquitous in the atmosphere, and it can exit in both
solid aerosol particles and fog and cloud droplets. Nitrate in the aqueous and particulate
phase can undergo photolysis to produce oxidizing active radicals, which will inevitably
affect various atmospheric chemical processes. However, the role of nitrate aerosols in
these atmospheric photochemical processes remains unclear. In this study, the effects
of nitrate photolysis on the aqueous phase oxidation of bisulfite under different
conditions were investigated. Results show that nitrate photolysis can significantly
promote the oxidation of bisulfite to sulfate. It is found that pH plays a significant role
in the reaction, and ammonium sulfate has significant impacts on regulating the pH of
solution and the enhancement of sulfate production. We also found an apparent
synergism among halogen chemistry, nitrate and its photochemistry and S(IV) aqueous
oxidation, especially the oxidation of halide ions by the nitrate photolysis and by the
intermediate peroxymonosulfuric acid ($HSO_5^-$) produced by the free radical chain
oxidation of S(IV) in acidic solution leads to the coupling of the redox cycle of halogen
with the oxidation of bisulfite, which promotes the continuous aqueous oxidation of
bisulfite and the formation of sulfate. In addition, it is also found that $O_2$ is of great
significance on nitrate photolysis for the conversion of $HSO_3^-$, and $H_2O_2$ generation
during the nitrate photolysis is verified. These results provide a new insight into the
heterogeneous aqueous phase oxidation pathways and mechanisms of $SO_2$ in cloud and
fog droplets and haze particles.



## ■ INTRODUCTION

The frequent haze events in China in recent years brought about severe impacts on
air quality, regional and global climates and human health (Huang et al., 2014; Ji et al.,
2014; Zheng et al., 2015b; Zheng et al., 2015a; Li et al., 2014). Secondary aerosols are
recognized as a main cause of particulate pollution during haze events (Guo et al., 2014),
of which secondary inorganic aerosols produced by the atmospheric conversions of $SO_2$
and $NO_x$ under adverse meteorological conditions are the predominant components (Li
et al., 2018b; Sun et al., 2014). Field measurements show that during haze episodes,
most of the particles especially the sub-micrometer particles were in the aqueous state
owing to the enrichment of aerosol liquid water by the elevated relative humidity (RH)
and inorganic fraction in particles (Liu et al., 2017b). As a result, the occurrence of
heterogeneous aqueous phase reaction has a marked tendency (Kong et al., 2018), and
the role of heterogeneous aqueous phase reaction of $SO_2$ played in secondary sulfate
aerosol formation was emphasized (Quan et al., 2015). Generally, the aqueous phase
oxidation pathways of $SO_2$ in the cloud and fog droplets include oxidation by $H_2O_2$,
oxidation by $O_3$ (Seinfeld and Pandis, 2006), and oxidation by $O_2$ under the catalysis of
certain transition metal ions (Ibusuki and Takeuchi, 1987). At pH ~2-5, the oxidation
by $H_2O_2$ has been considered as the dominant aqueous phase oxidation pathway for
sulfate formation (Ye et al., 2018; Shen et al., 2012). This pathway includes the uptake
of $SO_2$ (R1, see in Table 2) and subsequent aqueous phase oxidation of S(IV)
(Alexander et al., 2003). In the pH range of 2-7, most of the dissolved $SO_2$ will be
present as $HSO_3^-$ in the solution (Hua et al., 2008), and the oxidation of S(IV) species



leads to the formation of sulfate (R2). In addition, previous studies proposed that the
aqueous-phase oxidation of S(IV) by $NO_2$ played a dominant role in sulfate production
in northern China (Wang, 2016; Cheng et al., 2016), and the acid-base neutralization of
$NH_3$ is thought to be the reason of promoting $SO_2$ and $NO_2$ to form corresponding
sulfate and nitrate. However, their conclusion was doubted by Liu et al. and Guo et al.
(Liu et al., 2017a; Guo et al., 2017), who found that the reaction on the actual fine
particles with pH values of ~4.2 was too slow to account for sulfate formation.
Therefore, some potential oxidation pathways and mechanisms for atmospheric
aqueous phase oxidation of S(IV) still remain poorly understood.
Nitrate aerosol constitutes a substantial fraction of fine particles (Tao et al., 2018),
and surface nitrate can change the hygroscopicity of original particles (Hoffman et al.,
2004), which can be transformed into droplets at a certain RH. Field observation
showed that the remarkably enhanced nitrate formation was observed during haze
episodes in China (Ji et al., 2014; Tao et al., 2018), which is coincident with the high
level of sulfate (Zheng et al., 2015b; Quan et al., 2015), suggesting the impact of nitrate
on sulfate formation or the mutual influence between their formation processes in the
atmosphere (Du et al., 2019; Kong et al., 2014). Several studies have verified that nitrate
does participate in the heterogeneous conversion of $SO_2$. Du et al. (2019) proved that
the photolysis of adsorbed nitrate can be coupled with $SO_2$ oxidation and sulfate
formation, which is reflected in the consumption of adsorbed nitrate and the formation
of adsorbed $N_2O_4$ during the introduction of $SO_2$. Kong et al. (2014) found that nitrate
can accelerate the formation of sulfate on hematite and lead to the generation of surface-





adsorbed $HNO_3$ and gas-phase $N_2O$ and HONO under no light. And recently, Gen et al.
(2019b) proposed that nitrate photolysis at 250 nm is the major source of $NO_2$ and •OH
to oxidize dissolved $SO_2$, in which •OH is believed to be formed when $O_2$ exists, and
the reaction of •OH with dissolved $SO_2$ generates $SO_3^{2-}$, which leads to the chain
reaction involving $SO_5^-$, $HSO_5^-$ and $SO_4^-$ to produce multiple $SO_4^{2-}$ from each attack of
•OH on dissolved $SO_2$, providing a new insight for the heterogeneous aqueous phase
oxidation of $SO_2$ by nitrate photolysis. However, the impacts of various oxidizing
species produced and/or initiated by nitrate photolysis under different conditions on the
aqueous phase oxidation of $SO_2$ remain unclear. To further reveal the aqueous phase
oxidation mechanisms of $SO_2$ in ambient atmosphere, more potential pathways for
oxidant generation and S(IV) aqueous phase oxidation, and more ambient factors such
as the influence of coexisting substances, are still needed to be explored.
Halide ions (e.g. $Cl^-$, $Br^-$, $I^-$) are ubiquitous in the atmosphere via the transport of
sea salt aerosols, the emission of fossil fuel combustion and biomass burning (Richards
et al., 2011; Cahill et al., 1992; Cheng et al., 2000). The influence of halide ions on the
formation of $NO_2$ during the photochemistry of halide-nitrate ion mixtures has been
extensively studied (Richards et al., 2011; Richards and Finlayson-Pitts, 2012;
Richards-Henderson et al., 2013; Wingen et al., 2008; Custard et al., 2017). For example,
Wingen et al. (2008) observed enhanced $NO_2$ production from photolysis of
deliquesced nitrate aerosols containing chloride ions, and the reason was suggested that
halide ions can draw nitrate ions closer to the interface. Richards et al. (2011) found
that irradiated aqueous mixtures of $NaBr/NaNO_3$ exhibit an enhancement in the rates





of formation of $NO_2$ and $Br_2$ as the bromide mole fraction increased. However, up to
now, little attention has been paid to the effect of chemistry of halide ions on sulfate
formation under nitrate photolysis.

In this study, the aqueous phase oxidation of bisulfite influenced by nitrate

chemistry was investigated under different pH and irradiation intensity. Compared to
earlier works, different mechanisms for the generation of oxidants and the oxidation of
S(IV) species were discussed. The pH was controlled by added ammonium sulfate and
ammonium bisulfate because they coexist widely with nitrate in the atmospheric aerosol
particles, in which the crucial role of $NH_4^+$ on sulfate formation was also highlighted.
In the meantime, 2-propanol was premixed as inhibitor and •OH scavenger to examine
the roles of $O_2$ and •OH on the aqueous phase oxidation of bisulfite. Based on this study,
the formation of $H_2O_2$ via the recombination of •OH was verified. And furthermore, the
promotion effect of the redox cycling of halogen on the aqueous phase oxidation of
bisulfite under nitrate photolysis was investigated.

■    **MATERIALS AND METHODS**

**Materials.** Aqueous stock solutions of sodium bisulfite (ACS reagent, Sigma-

Aldrich), ammonium nitrate, ammonium sulfate (≥99.99% metals basis, Aladdin),
ammonium bisulfate (≥99.99% metals basis, Aladdin), sodium chloride (analytical
reagent, Shanghai Qiangshun Chemical Reagent Co., Ltd.), sodium bromide (≥99.99%
metals basis, Aladdin) and sodium iodide (≥99.99% metals basis, Aladdin) were
prepared by dissolving the corresponding salt in Milli-Q water. 2-propanol (HPLC Plus)



109 was purchased from Sigma-Aldrich. All the chemicals were used without further

110 purification.

111   **Photochemical Experiments.** Experiments were conducted in a cylindrical quartz

112 glass cell equipped with an inlet and outlet ports for gas transport and solution sampling,

113 respectively. The top of the cell was sealed with a quartz window (JGS2). Solutions

114 were irradiated from above by simulated sunlight using a Xenon lamp coupled with an

115 optical fiber (CEL-TCX250, Beijing China Education Au-light Co., Ltd.), and a water

116 filter was used in front of the radiation source. The reason is that both solar irradiation

117 and xenon lamp irradiation contain a small fraction (<5%) of UV light (Niu et al., 2013).

118 To further explore the reaction mechanisms, irradiation was also conducted by a high-

119 pressure mercury lamp (MERC500, NBet Technology Co., Ltd) coupled with a 313 nm

120 UV optical filter. And 8 mW/cm$^2$ (measured by an UV light meter, LS125-UVB,

121 Linshang Technology Co., Ltd) was selected as the experimental light intensity after

122 the pre-experiment, which could also achieve the expected results. A thermostatic water

123 bath circulating water through the reactor jacket was used to keep the temperature

124 constant at 25 °C.

125   **Measurements of SO$_4^{2-}$ and NO$_x$.** During the reaction, liquid samples were taken

126 out at given time intervals (0, 20, 40, …, 120 min) and 0.1 mL 2-propanol was added

127 to each sample to suppress the continuous oxidation by oxygen in the air during the test

128 process (Braga and Connick, 1982; Alyea and Bäckström, 1929). Then these samples

129 were analyzed by an ion chromatography (940 Professional IC Vario, Metrohm,

130 Switzerland), which was equipped with a separation column of Metrosep A supp 5-250



and a Metrosep A supp 5 S guard column for anion, and a Metrosep C 6-150 analytical
column and a Metrosep C 6 guard column for cation. 5% acetone was added to the
anion eluent to distinguish the peaks of sulfite and sulfate, so as to obtain the accurate
content of sulfate in the sample. Detection of $NO_x$ was conducted in the absence of
bisulfite. Nitrogen bubbling was used to help the produced $NO_x$ escaped from the
solution, and then $NO_x$ was measured using a $NO$-$NO_2$-$NO_x$ analyzer (42i, Thermo
Scientific, USA).

■    **RESULTS AND DISCUSSION**
**Aqueous Oxidation of Bisulfite by Nitrate Photolysis.** $NH_4NO_3$ and $(NH_4)_2SO_4$
are the major constituents of atmospheric particulate matter (Sun et al., 2013). Thus,
$NH_4NO_3$ was selected as the source of nitrate in this study (Shen et al., 2012). The
aqueous oxidation of bisulfite influenced by nitrate photolysis was carried out under
different conditions, as shown in Figure 1(a). The reaction with nitrate and light (S1,
13.58 $\mu M \cdot min^{-1}$) has a higher sulfate formation within 120 min than that in the dark (S2,
13.00 $\mu M \cdot min^{-1}$), verifying the promotion of nitrate photolysis on $HSO_3^-$ oxidation.
Considering the insignificant increase of sulfate yield, experiments were conducted
under 313 nm UV light (Figure 1(b)) again to explore the reaction mechanisms. Sulfate
formation remarkably enhanced in the presence of nitrate photolysis (S'1, 63.80
$\mu M \cdot min^{-1}$) compared with the low yield in dark. It may be attributed to the oxidation of
$HSO_3^-$ by various oxidizing species produced by nitrate photolysis, such as •OH (Gen
et al., 2019b; Mack and Bolton, 1999). •OH can be produced directly (R3) (Scharko et


al., 2014), or via indirect pathway induced by nitrate photolysis (R4-10) (Li et al., 2018b;
Yabushita et al., 2008; Ye et al., 2016; Bao et al., 2018): A simulated solar irradiation
of an aqueous solution of nitrate generates $O^-$ that readily reacts with $H^+$ to form •OH.
The secondary photolysis of HONO produced from nitrate photolysis (R6-R8) is also a
source of •OH, and the photo-formation of •OH from HONO and $NO_2^-$ is pH dependent
(Arakaki et al., 1999). It should be pointed out that the indirect pathways for •OH
formation processes include photoisomerization of nitrate to produce intermediate
peroxynitrite ($ONOO^-$) as well, which can combine with $H^+$ to form peroxynitrous acid
(HOONO). HOONO can produce •OH upon decomposition but $ONOO^-$ does not, and
hence a decrease of pH would enhance the yield of •OH photoproduction (Mack and
Bolton, 1999), which favors the oxidation of $HSO_3^-$. The contribution of •OH to sulfate
formation was discussed later. $NO_2$, another direct product of nitrate photolysis (R3 and
R4), is also a key factor that leads to $HSO_3^-$ oxidation. $HSO_3^-$ is oxidized by $NO_2$
directly (R11) or indirectly (R12) (Li et al., 2018b; Cheng et al., 2016; Guo et al., 2017;
Clifton et al., 1988; Gen et al., 2019a). The $NO_x$ analysis results in Figure 2 show that
$NO_2$ is generated rapidly and the yield reached the maximum (~ 54 ppb) within 20 min,
and then maintained at a relatively stable rate. The simultaneous formation of NO is
due to the photolysis of partial $NO_2$ and HONO (R13 and R14). Additionally, the
intermediate products such as nitrite ($NO_2^-$) and nitrous acid (HONO), i.e., N(III)
species are also considered as the contributors of aqueous phase $SO_2$ oxidation for
sulfate formation as well by previous studies (Gen et al., 2019a; Li et al., 2018b; Kong
et al., 2014). And even N(III) species was considered as the main contributors to the



heterogeneous $SO_2$ aqueous phase oxidation under 300 nm irradiation, followed by $NO_2$
contribution (Gen et al., 2019a). Therefore, the enhanced sulfate formation in this study
is a combined result of the oxidation of bisulfite by various oxidizing species produced
by the photolysis of nitrate.
The direct oxidation of $HSO_3^-$ by $O_2$ cannot be ignored according to S3 (8.74
$\mu M \cdot min^{-1}$). Meanwhile, compared to S'1, the low sulfate formation influenced by
nitrate photolysis under oxygen-free condition (S'4, 11.92 $\mu M \cdot min^{-1}$) indicates the key
role of oxygen for the process of nitrate photolysis affecting the conversion of $HSO_3^-$.
Details were discussed later. And no obvious sulfate generation in the absence of
ammonium nitrate and $O_2$ (S5) was observed. The small amount of sulfate increment is
attributed to the oxidation of $HSO_3^-$ by some dissolved $O_2$ in the solution. Another
interesting result was found via the comparison between S2 (13.00 $\mu M \cdot min^{-1}$) and S3
(8.74 $\mu M \cdot min^{-1}$), that is, nitrate itself can greatly promote the oxidation of bisulfite in
the solution under dark condition. This result is consistent with our previous study in
which nitrate facilitates the heterogeneous conversion of $SO_2$ on humid hematite
particles in the dark (Kong et al., 2014). Furthermore, this result also confirms our
previous finding that high-nitrate haze episodes favor the heterogeneous aqueous
oxidation of $SO_2$ and the formation of sulfate (Kong et al., 2018). Additionally, state-
of-the-art air quality models that rely on sulfate production mechanisms requiring
photochemical oxidants fail to predict the high levels of sulfate because the sunlight is
usually thought to be weak during haze events (Cheng et al., 2016), this may provide a
reasonable explanation for the enhanced conversion of atmospheric $SO_2$ and the



enhanced formation of secondary sulfate aerosols observed in severe haze episodes with
very weak radiation.

**Effect of pH on Sulfate Formation.** pH is an important factor affecting the aqueous
phase formation of sulfate (Barth et al., 2000), and then its effect on the reaction was
investigated. The pH of initial solution was 4.32 when the concentrations of $NaHSO_3$
and $NH_4NO_3$ were both 30 mM. Hence, the formation of sulfate discussed before
confirms the occurrence of nitrate photolysis under acidic conditions, as reported by
earlier works (Gen et al., 2019b; Scharko et al., 2014; Benedict et al., 2017).
Additionally, ammonium sulfate (AS), a salt of a strong acid and a weak base, is the
main form of atmospheric sulfate aerosol (Appel et al., 1978), and ammonium bisulfate
(ABS) has a stronger acidity, and thus they will inevitably affect the pH of various
aqueous phase system in the atmosphere, such as cloud and fog droplets. Therefore,
they were used to further adjust the expected pH value of the reaction solution in
different range in this study.
Figure 3(a) shows sulfate formation as a function of pH adjusted by the addition
of $(NH_4)_2SO_4$, $NH_4HSO_4$ and the mixture of $(NH_4)_2SO_4$ and $NH_4HSO_4$, respectively.
As can be seen from Figure 3(a), enhanced sulfate formation is observed in all
$(NH_4)_2SO_4$-adjusted systems, and more addition of $(NH_4)_2SO_4$ results in lower pH and
more significant sulfate formation. This result not only shows the important role of the
pH in sulfate formation, but also indicates a crucial role of ammonium sulfate in
regulating pH in the enhancement of sulfate formation. This is of great significance for
understanding the behavior of ammonium sulfate in the atmosphere. On the one hand,
the lower pH favors the formation of •OH as described by reactions R3-R10. On the
other hand, it may because that $NO_3^-$ under acidic conditions is more easily photolyzed
to produce •OH or HONO (Mack and Bolton, 1999; Bao et al., 2018; Turnipseed et al.,
1992). As a result, an increasingly enhanced sulfate formation is achieved as the pH
decreases. However, totally different results are obtained in $NH_4HSO_4$-adjusted systems.
The oxidation of bisulfite is greatly suppressed when pH < 2.08. One possible
explanation is that $HSO_3^-$ concentration is greatly reduced when pH < 2.08 due to the
equilibrium of reaction R1, even though nitrate photolysis under acidic conditions
continues, which will lead to the reduction of sulfate formation. This result may suggest
that there is an optimum pH value for sulfate formation. In addition, the middle range
of pH is reached by adding the mixed solution of $(NH_4)_2SO_4$ and $NH_4HSO_4$, and the
enhanced sulfate formation is also observed. Sulfate formation as a function of pH is
described in Fig 3(b). As can be seen from it, the pH for the highest sulfate formation
is about 3.86, which is within the pH range of atmospheric particles. Generally,
atmospheric $PM_{2.5}$ is acidic due to partial neutralization of acidic sulfate and nitrate
aerosols under some conditions (Guo et al., 2017; Liu et al., 2017). Liu et al. (2017)
found that fine particles were moderately acidic during severe haze episodes in northern
China, with a pH range of 3.0-4.9 and an average of 4.2. Guo et al. (2017) observed that
the $PM_1$ pH, regardless of ammonia levels, was always acidic, with an average of 4.2,
even for the unusually high $NH_3$ levels found in Beijing (pH = 4.5). Meanwhile,
although the sunlight is usually thought to be weak during haze events, field



observations showed that the photochemical reactivity during the winter or haze
episodes is still relatively high (Ye et al., 2018; Tan et al., 2018). Therefore, this result
suggests the new aqueous phase oxidation pathways coupled with nitrate photolysis,
which may play significant roles in the formation of secondary sulfate and the
occurrence and evolution of haze episodes in China.

Additionally, under the same conditions, the comparison of $NaNO_3$ as the source of

$NO_3^-$ for sulfate formation was carried out. Different conclusion from Gen et al. was
drew, who believed that the type of cation has little influence on nitrate photolysis (Gen
et al., 2019a). As can be seen from Fig 4, sulfate yields with $NaNO_3$ photolysis are both
lower than that with $NH_4NO_3$ at the same concentration, for that the hydrolysis of $NH_4^+$
may maintain a stable and low pH of the solution during the process, which is more
conducive to the formation of sulfate as described by R3-R10. Meanwhile, the same
low sulfate yields by photolysis of $NaNO_3$ and $NH_4NO_3$ at pH < 2 verify that too low
pH is unfavorable to sulfate formation again. To further investigate the role of $NH_4^+$,
the photolysis of $NO_3^-$ with different $NH_4^+$ content was conducted within the optimum
pH range. Results in Fig S1 show that sulfate yield increases with the increase of $NH_4^+$
content, revealing the significant role of $NH_4^+$ on the reaction. This may be because that
$NH_3 \cdot H_2O_{(aq)}$ and/or $NH_{3(aq)}$, which were produced by the hydrolysis of $NH_4^+$ (RS1), can
be oxidized by nitrate photolysis product $NO_2$ (RS2-RS5), and thus the shift of chemical
equilibrium enhances the photolysis of nitrate and the generation of •OH. Also, the
standard Gibbs energy of formation for the two reactions indicate that the oxidation
reactions of $NH_3 \cdot H_2O_{(aq)}$ and $NH_{3(aq)}$ by $NO_2$ are spontaneous ($\Delta_f G^\circ < 0$, listed in Text



S2). Therefore, the generation of •OH enhances simultaneously, thus leading to the
increased sulfate formation. In addition, it is reported that $NH_3$ can promote the
hydrolysis of $NO_2$ and induce the explosive growth of HONO via reaction R6 by
reducing the free energy barrier of the reaction and stabilizing the product state (Li et
al., 2018a; Xu et al., 2019). The HONO produced releases •OH upon photolysis (R10).
The highly oxidative free radicals promote the formation of sulfate. These results reflect
the crucial role and contribution of ambient widely-existing ammonium sulfate in the
formation of sulfate aerosols and photochemical pollution in the atmosphere.

**Role of $O_2$ and •OH in the aqueous oxidation of bisulfite.** As mentioned, $O_2$ and
•OH play important roles in the oxidation of $HSO_3^-$. We discussed it further here. It is
reported that alcohols can be used as inhibitors of sulfite and bisulfite oxidation (Braga
and Connick, 1982; Alyea and Bäckström, 1929). Braga and Connick found that the
oxidation of S(IV) by $O_2$ is a chain reaction, and alcohols can inhibit the reaction by
chain termination (Braga and Connick, 1982), but Alyea and Bäckström claimed that
the oxidation of the S(IV) induces the oxidation of alcohols and the alcohols can be
easily oxidized to aldehydes and ketones (Alyea and Bäckström, 1929). In addition, in
the presence of 2-propanol, •OH reacts primarily with 2-propanol ($k=1.9\times10^9$ $M^{-1}s^{-1}$)
through α-hydrogen abstraction generating 1-hydroxy-1-methylethyl radical, while at
lower concentrations of 2-propanol, the scavenging reaction of •OH with $H_2O_2$ (R15)
become competitive (Hislop and Bolton, 1999). The formation of $H_2O_2$ in the reaction
systems will be discussed later. Therefore, in this study, 2-propanol was added to





investigate the role of $O_2$ and •OH in the aqueous oxidation of bisulfite. The results are
presented in Fig 5. As can be seen from Fig.5, a slight reduction of sulfate formation
after the addition of 2-propanol was observed in the dark, indicating the suppression of
2-propanol on the oxidation of $HSO_3^-$ by $O_2$ because that 2-propanol can inhibit the
chain reaction between $HSO_3^-$ and $O_2$ by chain termination (Braga and Connick, 1982),
or be easily oxidized by $O_2$ induced by the oxidation of $HSO_3^-$ (Alyea and Bäckström,
1929), as mentioned before. The inconspicuous inhibition showed that $O_2$ oxidation is
not the main direct contributor on the aqueous phase oxidation of bisulfite in the
presence of nitrate.
However, as proved before, $O_2$ is of great significance on nitrate photolysis for the
conversion of $HSO_3^-$. On the one hand, the inhibition of 2-propanol on sulfate yield in
the presence and absence of $O_2$ under light were similar (Figure 5), demonstrating that
$O_2$ has little effect on the generation of •OH by nitrate photolysis. This is definitely
distinct with previous report by Gen et al. (2019b), who found that •OH is generated in
the presence of $O_2$. Furthermore, they claimed that the contribution of •OH pathway on
the oxidation of dissolved $SO_2$ was almost absent (<1%) under nitrate photolysis at 300
nm (Gen et al., 2019a). In our study, 8.59% and 25.02% of sulfate formation in the
presence and absence of $O_2$ respectively under 313 nm irradiation were owing to the
•OH oxidation, which is not obviously negligible. The quantum yield of •OH ($\Phi_{•OH}$)
was measured further (Text S4). The calculated $\Phi_{•OH}$ in Table 1 shows that lower pH
facilitates the generation of •OH during nitrate photolysis, verifying the points
discussed before. However, considering that the •OH would be recombined to



form $H_2O_2$ (discussed in next section) during its generation process, the calculated $\Phi_{\cdot OH}$
is not the completely accurate one.

In addition, it is reasonably inferred that $O_3$ was generated during nitrate photolysis

and became an important contributor for sulfate formation. As is known to all that in
aqueous solution $O(^3P)$ (one of the products of nitrate photolysis, R5) can react with
molecular oxygen to form ozone similar to the gas phase (R16) (Herrmann, 2007).
Therefore, it is expected that $HSO_3^-$ will be oxidized by $O_3$, but this still needs further
study.

**Generation of hydrogen peroxide during the photolysis process**

Previous studies haven't detected $H_2O_2$ during steady-state irradiation of $NO_2^-$ and

$NO_3^-$ solutions at $\lambda > 200$ nm (Daniels et al., 1968; Shuali et al., 1969; Mark et al., 1996),
but Wagner et al. once found $H_2O_2$ formation in flash photolysis of nitrate ions in acidic
aqueous solution (Wagner et al., 1980). However, Mack et al. thought that the
combination reaction between two •OH produced by nitrate photolysis is highly
unlikely due to the very low concentration and short lifetime of •OH, and they attributed
the formation of $H_2O_2$ observed by Wagner et al. to the $H_2O$ photolysis at $\lambda > 180$ nm
(Mack and Bolton, 1999). However, Yabushita et al. (2008) once again found that the
produced •OH by the photolysis of nitrate originated from the adsorption of nitric acid
can recombine to form $H_2O_2$ under low-temperature ice conditions.

$H_2O_2$ generation during the photochemical process of nitrate was verified in this

study. $H_2O_2$ was measured by titanium (III) sulfate spectrophotometry and the results



are depicted in Fig S2. The production of $H_2O_2$ shows a trend of increasing first and
then decreasing, which is due to the fact that $H_2O_2$ is photodegraded as its formation.
Considering the contributions of $NH_4^+$ and pH, and the absence of precursor of
hydroperoxyl radical ($HO_2$) in our reaction system, we can speculate that the formation
of $H_2O_2$ is owing to the recombination of •OH originated from nitrate photolysis and it
may make an important contribution to $HSO_3^-$ oxidation and sulfate formation. When
25 mM $(NH_4)_2SO_4$ and $NH_4HSO_4$ added, the pH of solution decreased from 5.00 to
4.63 and 0.97 respectively, as a result, $H_2O_2$ formation is found to be remarkably pH-
dependent. The formation of $H_2O_2$ displayed a higher efficiency at a lower pH, which
indicates that lower pH favors the photolysis of nitrate to generate •OH and the
formation of $H_2O_2$. These may reveal the new formation pathways to the source of $H_2O_2$
in cloud, fogs and liquid haze particles, which may be important for the conversion of
atmospheric $SO_2$ and the formation of atmospheric sulfate, as well as the occurrence
and evolution of haze events. Such aspects need to be further explored in future.

**Effect of light intensity on aqueous phase formation of sulfate.**
Although sunlight is dimmed by ambient PM during haze events, it is not completely
absent (Xia et al., 2018), and the active photochemistry is also found during the winter
or haze episodes and facilitates the production of secondary pollutants (Lu et al., 2019).
In the late period of haze events with weak solar radiation, significant increase of sulfate
and nitrate was observed (Quan et al., 2015). In order to reveal the impact of the solar
radiation on sulfate formation through the aqueous phase pathways we mentioned



above, we carried out experiments under a higher light intensity. As shown in Fig S5, a
higher light intensity more favors sulfate formation than that under 8 mW/cm$^2$
irradiation.

**Role of Halide Ions in Aqueous Phase Oxidation of Bisulfite.** Field observations
indicate that nitrate ions commonly coexist with halide ions in aged sea salt particles
and atmospheric particles due to the transport of sea salt aerosols, the emission of fossil
fuel combustion and biomass burning. Our previous study verified that high-nitrate
haze episodes occurred in Shanghai (a coastal mega-city) favored heterogeneous
aqueous oxidation of $SO_2$ (Kong et al., 2018). However, little is known about the role
of halide ions in the heterogeneous aqueous oxidation of $SO_2$ and the formation of
sulfate. Therefore, the role of halide ions in the aqueous phase oxidation of $HSO_3^-$ under
nitrate photolysis was further investigated. Results in Fig 7(a) show that sulfate
formation is evidently enhanced in the presence of halide ions $X^-$ or $Y^-$ (X, Y: Cl, Br
and I), indicating the promotion effect of halide ions on aqueous phase oxidation of
bisulfite in the presence of nitrate and light.
In this study, halide-induced enhancement of nitrate photolysis for sulfate formation
owing to halide photochemistry and the redox cycle of halogen was proposed for the
first time. Firstly, direct photolysis of $X^-$ generates $X^•$, followed by the rapid reaction
of $X^•$ with halide ion $X^-$ (or $Y^-$) to form $X_2^{•-}$ (or $XY^{•-}$) as reaction R17 and R18 (Zhang
and Parker, 2018). Secondly, halide ions can be oxidized by nonhalogen radicals from
nitrate photolysis and the free radical chain oxidation of S(IV) [e.g. $^•OH$ (R19); sulfate



radical, $SO_4^{\cdot-}$ (R21)] to generate halogen radicals, and halogen radicals are present at
higher concentrations relative to •OH in more acidic solutions (Zhang and Parker, 2018),
as indicated by reaction R19-R22. Additionally, peroxymonosulfuric acid ($HSO_5^-$)
produced by the free radical chain oxidation of S(IV) is known to react with halide ions
to produce HOX (Mozurkewich, 1995), e.g. reaction R23, and then HOX photolysis
generates $X^{\cdot}$ and •OH (reaction R24). Nonradical halogen oxidant products $X_2$ and
HOX can also be formed from rapid reactions R25-R29 (Zhang and Parker, 2018). And,
the reaction of HOBr with $Br^-$ (or $Cl^-$) can also lead to the production of $Br_2$ (or BrCl),
as described by R30 and R31 (Mozurkewich, 1995; Vogt et al., 1996). Moreover, •OH
can oxidize both $Cl^-$ and $Br^-$ to form photochemically active halogen molecules such as
$Br_2$, BrCl and $Cl_2$ (R32-R34) (Alexander et al., 2003; Richards and Finlayson-Pitts,
2012; Richards-Henderson et al., 2013). In addition, $Br^-$ can be oxidized to $Br_2$ by nitrate
under acidic conditions in the dark (R35), and the formed HONO further oxidizes $Br^-$
(Richards-Henderson et al., 2013). These nonradical halogen oxidant products may
contribute to subsequent oxidation of $HSO_3^-$ (R36, R37), e.g., rapid S(IV) oxidation by
HOCl and HOBr (R38, R39) (Richards and Finlayson-Pitts, 2012; Vogt et al., 1996). It
should be noted that althrough the direct photolysis of halide ions exists, this pathway
may play a small role in our study because the used xenon lamp emits very few specific
wavelengths for the reaction.
Therefore, in this study, compared to that in the absence of halide ions, extra
oxidations caused by the introduction of halide ions significantly promote the
conversion of $HSO_3^-$ and the formation of sulfate, showing an apparent synergism


among halogen chemistry, nitrate photolysis and S(IV) aqueous oxidation. Herein, it
should point out that the redox cycle of halogen for the oxidation of $HSO_3^-$ under nitrate
photolysis exists. That is, the oxidants HOX and $X_2$ produced by the oxidation of halide
ions react with bisulfite to generate sulfate as well as to regenerate halide ions, thus
forming a redox cycle of halogen to achieve the continuous oxidation conversion of
bisulfite. Halide ions act as catalysts in the cyclic reactions. Furthermore, Fig 7(a) also
shows that the sulfate yields of the three groups with the addition of $Cl^-$, $Br^-$ and $I^-$
exhibit an increasing tendency, especially the one with added $I^-$, which is much higher
than that of the $Cl^-$ and $Br^-$. On the one hand, pH plays a key role. The pH of bulk
solution with addition of $Cl^-$, $Br^-$ and $I^-$ are 4.42, 4.20 and 3.60, respectively. As depicted
in Fig 3(b) the lower pH of the three favors sulfate formation. On the other hand, sulfate
yields present an order of $Cl^-$-added system<$Br^-$-added system<$I^-$-added system, which
is consistent with their reducibility. Compared with $Cl_2/Cl^-$ and $Br_2/Br^-$ redox cycles,
the formed $I_2$ has stronger oxidizability, and $I^-$ has stronger reducibility, and thus it is
easier to form $I_2/I^-$ (or $I_3^-/I^-$) redox cycle in our reaction system, resulting in a higher
sulfate yield. In addition, it is noteworthy that the added halide ions promote the
aqueous phase oxidation of $HSO_3^-$ in the dark as well, as can be seen from Fig 7(b). But
the sulfate yield of dark reaction with different $X^-$ is lower than that with light. This
result indicates the simultaneous contributions of the decreased pH and the redox cycle
of halogen originated from the reaction R22, R23 and R35 and the aqueous phase
oxidation of $HSO_3^-$ in the absence of light. Similarly, the autocatalytic reaction of this
system is achieved as well in the dark.





The results suggest that the redox cycle of halogen caused by the nitrate and its
photochemistry as well as $HSO_5^-$ will be coupled with the aqueous phase oxidation
process of $SO_2$ to greatly promote the formation of sulfate, both under the light and dark
conditions, which is of great significance to understand the impacts of sea salt aerosol
in coastal areas of China and the halide ions from coal combustion in northern areas of
China on the formation of secondary aerosols and the occurrence and evolution of haze
episodes or air pollution in China. Additionally, the transport of biomass burning can
induce air pollution and haze episodes in the receptor areas by the significantly
enhanced formation of secondary sulfate and nitrate aerosols (Du et al., 2011; Tong et
al., 2020). The interactions between the plume of biomass burning and local pollutants
are often used to explain the enhanced secondary aerosol formation, but the real reason
remain unclear. Therefore, the role of halogen chemistry we studied may provide a
reasonable explanation for the occurrence of haze episodes induced by biomass burning.
These lay a foundation for us to extend the relevant research to the formation of
atmospheric SOA and organic halides in the future.

**CONLUSIONS AND ATMOSPHERIC IMPLICATIONS**
The effects of nitrate photolysis on the aqueous phase oxidation of bisulfite under
different conditions have been investigated. A combining contribution of various
oxidizing species (•OH, $NO_2$, $NO_2^-$ and HONO, etc.) produced via the photolysis of
nitrate achieves the enhanced oxidation of bisulfite and the enhanced sulfate formation.
pH plays a significant role in bisulfite oxidation and sulfate formation. The highest


sulfate formation occurs in the range of moderate acidity, about 3.86, which is within
the pH range of atmospheric particles. Furthermore, $NH_4^+$ promotes the generation of
•OH and thus enhances the formation of sulfate, which is attributed to the oxidation of
$NH_4^+$ hydrolysis products $NH_3•H_2O_{(aq)}$ and $NH_{3(aq)}$ by $NO_2$, and the promotion role of
$NH_4^+$ hydrolysis products in the hydrolysis of $NO_2$ to form HONO. Additionally, $O_2$
remarkably facilitates nitrate photolysis for the conversion of $HSO_3^-$. And $H_2O_2$ can be
generated via the recombination of •OH from nitrate photolysis. More importantly, the
redox cycle of halogen coupled with S(IV) oxidation significantly promote the
conversion of $HSO_3^-$ and the formation of sulfate under nitrate photolysis or in acidic
nitrate solution in the dark, showing the apparent synergism among halogen chemistry,
nitrate chemistry and S(IV) aqueous oxidation. Our study verifies that S(IV) oxidation
can be coupled not only with nitrate and its photochemistry, but also with the redox
cycle of halogen, which greatly promotes the formation of sulfate.

Results from this study have important atmospheric implications. Firstly, in recent

years in China, photochemical pollution and haze episodes have occurred more
frequently and the haze episodes with high nitrate level are increasingly apparent. Due
to the relatively high RH and high concentrations of particulate nitrate, the aerosol is
mostly in the aqueous phase during haze episodes (Liu et al., 2017b; Kong et al., 2018;
Lu et al., 2019). Meanwhile, sunlight is not completely absent during haze episodes (Ye
et al., 2018; Xia et al., 2018), and the active photochemistry is also found during the
winter or haze episodes and facilitates the production of secondary pollutants (Lu et al.,
2019). Therefore, the enhanced aqueous phase oxidation of bisulfite and the enhanced

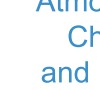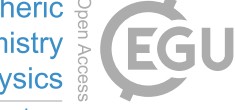

formation of sulfate under the coupling with nitrate photolysis or in the nitrate solution
under dark condition may occur in the haze episodes and promote the occurrence and
evolution of haze episodes in China. In the meantime, the important roles of ammonium
sulfate in regulating the pH of solution and the enhancement of sulfate production will
also be reflected.
Secondly, nitrate aerosol is ubiquitous in the atmosphere, and it will inevitably mix
with halide ions ($X^- = Cl^-$, $Br^-$, $I^-$) from sea salt particles, coal combustion and biomass
burning, especially over the coastal areas and the northern areas of China. The typical
haze areas in China include Beijing-Tianjin-Hebei area, Yangtze River Delta area and
Pearl River Delta area, the three coastal areas in nature, which may suggest the potential
roles of halide ions in the occurrence and evolution of hazes. Therefore, the reactions
such as those reported here may occur in aqueous phase aerosols (e.g. fog and cloud
droplets) or aerosol particles with water film, and the formation of $X\cdot$, $NO_2$, $HOX$, $X_2$,
$XY$ or the $X_2/X^-$ redox cycle will further enhance the conversion of $SO_2$ and/or
oxidation of organic gaseous precursors, thus promoting the occurrence and evolution
of air pollution or haze. Our study highlights the roles of halide ions in the formation
of secondary aerosols and air pollution. Meanwhile, the formed $X\cdot$, $X_2$ or $XY$ may be
liberated from the aqueous phase particles or the particles with water film into the air,
and thus they will affect the tropospheric $O_3$ level.
Finally, secondary organic aerosol (SOA) is also an important component of
atmospheric aerosols, and it has attracted much attention all the time. Our investigation
provides a case study on the formation of secondary inorganic aerosols, which can be



extended to the study on the influences of nitrate photochemistry and the coupled redox
cycle of halogen on the formation of SOA, such as the hydroxylation and halogenation
of organic substrates, etc.
■ **ASSOCIATED CONTENT**
**Supporting Information**
■ **AUTHOR INFORMATION**
**Corresponding Author**
*E-mail: ldkong@fudan.edu.cn
**ORCID**
**Notes**
The authors declare no competing financial interest.
■ **AUTHOR CONTRIBUTION**
Lingdong Kong, as a tutor, guided and provided suggestions in the whole research
process. Lu Chen undertook the main experiment and prepared the manuscript with
contributions from all co-authors. Songying Tong and Kejing Yang provided guidance
on experimental instruments and their operation methods. Shengyan Jin and Chao Wang
carried out some basic experiments. And Lin Wang provided a part of financial support
on instruments and materials.
■ **ACKNOWLEDGEMENTS**
This study was supported by the National Natural Science Foundation of China
(Grant Nos. 21777027 and 21976032) and the National Key R & D Program of China
(2017YFC0209505).



■ **REFERENCES**
Alexander, L., Gaspar, D. J., Weihong, W., Hunt, S. W., Cowin, J. P., Colson, S. D., and Finlayson-Pitts,
B. J.: Reactions at interfaces as a source of sulfate formation in sea-salt particles, Science, 301, 340-
344, https://doi.org/10.1126/science.1085374, 2003.
Alyea, H. N., and Bäckström, H. L. J.: The inhibitive action of alcohols on the oxidation of sodium sulfite,
J. Am. Chem. Soc, 51, 90-109, https://doi.org/10.1021/ja01376a011, 1929.
Appel, B. R., Kothny, E. L., Hoffer, E. M., Hidy, G. M., and Wesolowski, J. J.: Sulfate and nitrate data
from the California Aerosol Characterization Experiment (ACHEX), Environ. Sci. Technol, 12,
418-425, https://doi.org/10.1021/es60140a005, 1978.
Arakaki, T., Miyake, T., Hirakawa, T., and Sakugawa, H.: pH dependent photoformation of hydroxyl
radical and absorbance of aqueous-phase N(III) ($HNO_2$ and $NO_2^-$), Environ. Sci. Technol, 33, 2561-
2565, https://doi.org/10.1021/es980762i, 1999.
Bao, F., Li, M., Zhang, Y., Chen, C., and Zhao, J.: Photochemical aging of Beijing urban PM2.5: HONO
production, Environ. Sci. Technol, 52, 6309-6316, https://doi.org/10.1021/acs.est.8b00538, 2018.
Barth, M. C., Rasch, P. J., Kiehl, J. T., Benkovitz, C. M., and Schwartz, S. E.: Sulfur chemistry in the
National Center for Atmospheric Research Community Climate Model: Description, evaluation,
features, and sensitivity to aqueous chemistry, J. Geophys. Res. Atmos, 105, 1387-1415,
http://dx.doi.org/10.1029/1999JD900773, 2000.
Benedict, K. B., McFall, A. S., and Anastasio, C.: Quantum yield of nitrite from the photolysis of aqueous
nitrate above 300 nm, Environ. Sci. Technol, 51, 4387-4395,
https://doi.org/10.1021/acs.est.6b06370, 2017.
Braga, T. G., and Connick, R. E.: Kinetics of the oxidation of bisulfite ion by oxygen, in: Flue Gas
Desulfurization, ACS Symposium Series, 188, American Chemical Society, 153-171,
http://dx.doi.org/10.1021/bk-1982-0188.ch008, 1982.
Cahill, T. A., Wilkinson, K., and Schnell, R.: Composition analyses of size-resolved aerosol samples
taken from aircraft downwind of Kuwait, spring 1991, J. Geophys. Res. Atmos, 97, 14513-14520,
http://dx.doi.org/10.1029/92JD01373, 1992.
Cheng, Y., Zheng, G., Wei, C., Mu, Q., Zheng, B., Wang, Z., Gao, M., Zhang, Q., He, K., Carmichael,
G., Pöschl, U., and Su, H.: Reactive nitrogen chemistry in aerosol water as a source of sulfate during
haze events in China, Sci Adv, 2, http://dx.doi.org/e1601530-e1601530, 10.1126/sciadv.1601530,
2016.

Cheng, Z. L., Chan, L. W. T., Cheng, K. K., and Lam, K. S.: Chemical characteristics of aerosols at
coastal station in Hong Kong. I. Seasonal variation of major ions, halogens and mineral dusts
between 1995 and 1996, Atmos. Environ, 34, 2771-2783, http://dx.doi.org/10.1016/S1352-
2310(99)00343-X, 2000.
Clifton, C. L., Altstein, N., and Huie, R. E.: Rate constant for the reaction of nitrogen dioxide with
sulfur(IV) over the pH range 5.3-13, Environ. Sci. Technol, 22, 586-589,
http://dx.doi.org/10.1021/es00170a018, 1988.
Custard, K. D., Raso, A. R. W., Shepson, P. B., Staebler, R. M., and Pratt, K. A.: Production and release
of molecular bromine and chlorine from the arctic coastal snowpack, ACS. Earth. Space. Chem, 1,
142-151, http://dx.doi.org/10.1021/acsearthspacechem.7b00014, 2017.
Daniels, M., Meyers, R. V., and Belardo, E. V.: Photochemistry of the aqueous nitrate system. I.
Excitation in the 300-m.mu. band, J. Phys. Chem, 72, https://doi.org/10.1021/j100848a002, 1968.



Du, C., Kong, L., Zhanzakova, A., Tong, S., Yang, X., Wang, L., Fu, H., Cheng, T., Chen, J., and Zhang,
S.: Impact of adsorbed nitrate on the heterogeneous conversion of $SO_2$ on alpha-$Fe_2O_3$ in the absence
and presence of simulated solar irradiation, Sci. Total. Environ, 649, 1393-1402,
http://dx.doi.org/10.1016/j.scitotenv.2018.08.295, 2019.
Du, H., Kong, L., Cheng, T., Chen, J., Du, J., Li, L., Xia, X., Leng, C., and Huang, G.: Insights into
summertime haze pollution events over Shanghai based on online water-soluble ionic composition
of aerosols, Atmos. Environ, 45, 5131-5137, http://dx.doi.org/10.1016/j.atmosenv.2011.06.027,
2011.
Dubowski, Y., Colussi, A. J., and Hoffmann, M. R.: Nitrogen dioxide release in the 302 nm band
photolysis of spray-frozen aqueous nitrate solutions. Atmospheric implications, J. Phys. Chem. A,
105, 4928-4932, http://dx.doi.org/10.1021/jp0042009, 2001.
Gen, M., Zhang, R., Huang, D. D., Li, Y., and Chan, C. K.: Heterogeneous oxidation of $SO_2$ in sulfate
production during nitrate photolysis at 300 nm: Effect of pH, relative humidity, irradiation intensity,
and the presence of organic compounds, Environ. Sci. Technol, 53, 8757-8766,
http://dx.doi.org/10.1021/acs.est.9b01623, 2019a.
Gen, M., Zhang, R., Huang, D. D., Li, Y., and Chan, C. K.: Heterogeneous $SO_2$ oxidation in sulfate
formation by photolysis of particulate nitrate, Environ. Sci. Technol. Lett, 6, 86-91,
http://dx.doi.org/10.1021/acs.estlett.8b00681, 2019b.
Gligorovski, S., Strekowski, R., Barbati, S., and Vione, D. J. C. R.: Environmental implications of
hydroxyl radicals (•OH), Chem. Rev, 115, 13051-13092, http://dx.doi.org/10.1021/cr500310b, 2015.
Guo, H., Weber, R. J., and Nenes, A.: High levels of ammonia do not raise fine particle pH sufficiently
to yield nitrogen oxide-dominated sulfate production, Sci. Rep, 7, 12109,
http://dx.doi.org/10.1038/s41598-017-11704-0, 2017.
Guo, S., Hu, M., Levy Zamora, M., Peng, J., Shang, D., Zheng, J., Du, Z., Wu, Z., Shao, M., Zeng, L.,
Molina, M., and Zhang, R.: Elucidating severe urban haze formation in China, P. Natl. Acad. Sci.
USA, http://dx.doi.org/111, 10.1073/pnas.1419604111, 2014.
Herrmann, H.: On the photolysis of simple anions and neutral molecules as sources of $O^-/OH$, $SO_x^-$ and
Cl in aqueous solution, Phys. Chem. Chem. Phys, 9, 3935-3964,
http://dx.doi.org/10.1039/b618565g, 2007.
Hislop, K. A., and Bolton, J. R.: The photochemical generation of hydroxyl radicals in the UV-
vis/Ferrioxalate/$H_2O_2$ system, Environ. Sci. Technol, 33, 3119-3126,
http://dx.doi.org/10.1021/es9810134, 1999.
Hoffman, R. C., Laskin, A., and Finlayson-Pitts, B. J.: Sodium nitrate particles: physical and chemical
properties during hydration and dehydration, and implications for aged sea salt aerosols, J. Aerosol.
Sci, 35, 869-887, http://dx.doi.org/10.1016/j.jaerosci.2004.02.003, 2004.
Hua, W., Chen, Z., Jie, C., Kondo, Y., Hofzumahaus, A., Takegawa, N., Lu, K., Miyazaki, Y., Kita, K.,
and Wang, H.: Atmospheric hydrogen peroxide and organic hydroperoxides during PRIDE-PRD'06,
China: their concentration, formation mechanism and contribution to secondary aerosols, Atmos.
Chem. Phys, 8, 6755-6773, https://doi.org/10.5194/acp-8-6755-2008, 2008.
Huang, R. J., Zhang, Y., Bozzetti, C., Ho, K. F., Cao, J. J., Han, Y., Daellenbach, K. R., Slowik, J. G.,
Platt, S. M., Canonaco, F., Zotter, P., Wolf, R., Pieber, S. M., Bruns, E. A., Crippa, M., Ciarelli, G.,
Piazzalunga, A., Schwikowski, M., Abbaszade, G., Schnelle-Kreis, J., Zimmermann, R., An, Z.,
Szidat, S., Baltensperger, U., Haddad, I. E., and Prévôt, A. S. H.: High secondary aerosol
contribution to particulate pollution during haze events in China, Nature, 514, 218,



https://doi.org/10.1038/nature13774, 2014.
Ibusuki, T., and Takeuchi, K.: Sulfur dioxide oxidation by oxygen catalyzed by mixtures of manganese
(II) and iron (III) in aqueous solutions at environmental reaction conditions, Atmos. Environ, 21,
1555-1560, http://dx.doi.org/10.1016/0004-6981(87)90317-9, 1987.
Ji, D., Li, L., Wang, Y., Zhang, J., Cheng, M., Sun, Y., Liu, Z., Wang, L., Tang, G., Hu, B., Chao, N., Wen,
T., and Miao, H.: The heaviest particulate air-pollution episodes occurred in northern China in
January, 2013: Insights gained from observation, Atmos. Environ, 92, 546-556,
http://dx.doi.org/10.1016/j.atmosenv.2014.04.048, 2014.
Kong, L., Du, C., Zhanzakova, A., Cheng, T., Yang, X., Wang, L., Fu, H., Chen, J., and Zhang, S.: Trends
in heterogeneous aqueous reaction in continuous haze episodes in suburban Shanghai: An in-depth
case study, Sci. Total. Environ, 634, 1192-1204, http://dx.doi.org/10.1016/j.scitotenv.2018.04.086,

2018.

Kong, L. D., Zhao, X., Sun, Z. Y., Yang, Y. W., Fu, H. B., Zhang, S. C., Cheng, T. T., Yang, X., Wang, L.,
and M., C. J.: The effects of nitrate on the heterogeneous uptake of sulfur dioxide on hematite,
Atmos. Chem. Phys., 14, 9451–9467, http://dx.doi.org/10.5194/acp-14-9451-2014, 2014.
Li, L., Huang, C., Huang, H., Wang, Y., Yan, R., Zhang, G., Zhou, M., Lou, S., Tao, S., and Wang, H. J.
A. e.: An integrated process rate analysis of a regional fine particulate matter episode over Yangtze
River Delta in 2010, Atmos. Environ, 91, 60-70, http://dx.doi.org/10.1016/j.atmosenv.2014.03.053,

2014.

Li, L., Duan, Z., Li, H., Zhu, C., Henkelman, G., Francisco, J. S., and Zeng, X. C.: Formation of HONO
from the $NH_3$-promoted hydrolysis of $NO_2$ dimers in the atmosphere, Proc. Natl. Acad. Sci. USA,
115, 7236-7241, http://dx.doi.org/10.1073/pnas.1807719115, 2018a.
Li, L., Hoffmann, M. R., and Colussi, A. J.: Role of nitrogen dioxide in the production of sulfate during
Chinese haze-aerosol episodes, Environ. Sci. Technol, 52, 2686-2693,
http://dx.doi.org/10.1021/acs.est.7b05222, 2018b.
Liu, M., Song, Y., Zhou, T., Xu, Z., Yan, C., Zheng, M., Wu, Z., Hu, M., Wu, Y., and Zhu, T.: Fine particle
pH during severe haze episodes in northern China, Geophys. Res. Lett, 44, 5213-5221,
http://dx.doi.org/10.1002/2017gl073210, 2017a.
Liu, Y., Wu, Z., Wang, Y., Xiao, Y., Gu, F., Zheng, J., Tan, T., Shang, D., Wu, Y., Zeng, L., Hu, M., P.
Bateman, A., and Martin, S.: Submicrometer particles are in the liquid state during heavy haze
episodes in the urban atmosphere of Beijing, China, Environ. Sci. Technol. Lett, 4,
http://dx.doi.org/10.1021/acs.estlett.7b00352, 2017b.
Lu, K., Fuchs, H., Hofzumahaus, A., Tan, Z., Wang, H., Zhang, L., Schmitt, S. H., Rohrer, F., Bohn, B.,
Broch, S., Dong, H., Gkatzelis, G. I., Hohaus, T., Holland, F., Li, X., Liu, Y., Liu, Y., Ma, X., Novelli,
A., Schlag, P., Shao, M., Wu, Y., Wu, Z., Zeng, L., Hu, M., Kiendler-Scharr, A., Wahner, A., and
Zhang, Y.: Fast photochemistry in wintertime haze: Consequences for pollution mitigation strategies,
Environ. Sci. Technol, 53, 10676-10684, http://dx.doi.org/10.1021/acs.est.9b02422, 2019.
Mack, J., and Bolton, J. R.: Photochemistry of nitrite and nitrate in aqueous solution: a review, J.
Photochem. Photobio. A, 128, 1-13, https://doi.org/10.1016/S1010-6030(99)00155-0, 1999.
Mark, G., Korth, H.G., Schuchmann, H.P., and von Sonntag, C.: The photochemistry of aqueous nitrate
ion revisited, J. Photochem. Photobio. A, 101, 89-103, https://doi.org/10.1016/S1010-

6030(96)04391-2, 1996.

Mozurkewich, M.: Mechanisms for the release of halogens from sea-salt particles by free radical
reactions, J. Geophys. Res, 100, 14199-14207, http://dx.doi.org/10.1029/94JD00358, 1995.



Niu, J., Li, Y., and Wang, W.: Light-source-dependent role of nitrate and humic acid in tetracycline
photolysis: kinetics and mechanism, Chemosphere, 92, 1423-1429,
http://dx.doi.org/10.1016/j.chemosphere.2013.03.049, 2013.

Quan, J., Liu, Q., Li, X., Gao, Y., Jia, X., Sheng, J., and Liu, Y.: Effect of heterogeneous aqueous reactions
on the secondary formation of inorganic aerosols during haze events, Atmos. Environ, 122, 306-312,
http://dx.doi.org/10.1016/j.atmosenv.2015.09.068, 2015.

Richards-Henderson, N. K., Callahan, K. M., Nissenson, P., Nishino, N., Tobias, D. J., and Finlayson-
Pitts, B. J.: Production of gas phase $NO_2$ and halogens from the photolysis of thin water films
containing nitrate, chloride and bromide ions at room temperature, Phys. Chem. Chem. Phys, 15,
17636-17646, http://dx.doi.org/10.1039/C3CP52956H, 2013.

Richards, N. K., Wingen, L. M., Callahan, K. M., Nishino, N., Kleinman, M. T., Tobias, D. J., and
Finlayson-Pitts, B. J.: Nitrate ion photolysis in thin water films in the presence of bromide ions, J.
Phys. Chem. A, 115, 5810-5821, http://dx.doi.org/10.1021/jp109560j, 2011.

Richards, N. K., and Finlayson-Pitts, B. J.: Production of gas phase $NO_2$ and halogens from the
photochemical oxidation of aqueous mixtures of sea salt and nitrate ions at room temperature,
Environ. Sci. Technol, 46, 10447-10454, http://dx.doi.org/10.1021/es300607c, 2012.

Scharko, N. K., Berke, A. E., and Raff, J. D.: Release of nitrous acid and nitrogen dioxide from nitrate
photolysis in acidic aqueous solutions, Environ. Sci. Technol, 48, 11991-12001,
http://dx.doi.org/10.1021/es503088x, 2014.

Seinfeld, J., and Pandis, S.: Atmospheric chemistry and physics: from air pollution to climate change,
John. Wiley. Sons, New York, 2006.

Shen, X., Lee, T., Guo, J., Wang, X., Li, P., Xu, P., Wang, Y., Ren, Y., Wang, W., Wang, T., Li, Y., Carn,
S. A., and Collett, J. L.: Aqueous phase sulfate production in clouds in eastern China, Atmos.
Environ,    62, 502-511, https://doi.org/10.1016/j.atmosenv.2012.07.079, 2012.

Shuali, U., Ottolenghi, M., Rabani, J., and Yelin, Z.: Photochemistry of aqueous nitrate solutions excited
in the 195-nm band, J. Phys. Chem, 73, 3445-3451, https://doi.org/10.1021/j100844a052, 1969.

Sun, Y., Jiang, Q., Wang, Z., Fu, P., Li, J., Yang, T., and Yin, Y.: Investigation of the sources and evolution
processes of severe haze pollution in Beijing in January 2013, J. Geophys. Res. Atmos, 119, 4380-
4398, https://doi.org/10.1002/2014JD021641, 2014.

Sun, Y. L., Wang, Z. F., Fu, P. Q., Yang, T., Jiang, Q., Dong, H. B., Li, J., and Jia, J. J.: Aerosol
composition, sources and processes during wintertime in Beijing, China, Atmos. Chem. Phys., 13,
4577-4592, https://doi.org/10.5194/acp-13-4577-2013, 2013.

Tan, Z., Rohrer, F., Lu, K., Ma, X., Bohn, B., Broch, S., Dong, H., Fuchs, H., Gkatzelis, G. I.,
Hofzumahaus, A., Holland, F., Li, X., Liu, Y., Liu, Y. H., Novelli, A., Shao, M., Wang, H. C., Wu,
Y. S., Zeng, L. M., H, M., Kiendler-Scharr, A., Wahner, A., and Zhang, Y. H.: Wintertime
photochemistry in Beijing: observations of $RO_x$ radical concentrations in the North China Plain
during the BEST-ONE campaign, Atmos. Chem. Phys., 18, 12391-12411,
https://doi.org/10.5194/acp-18-12391-2018, 2018.

Tao, J., Zhang, Z., Tan, H., Zhang, L., Wu, Y., Sun, J., Che, H., Cao, J., Cheng, P., Chen, L., and Zhang,
R.: Observational evidence of cloud processes contributing to daytime elevated nitrate in an urban
atmosphere, Atmos. Environ, 186, 209-215, https://doi.org/10.1016/j.atmosenv.2018.05.040, 2018.

Tong, S. Y., Kong, L. D., Yang, K. J., Shen, J. D., Chen, L., Jin, S. Y., Wang, C., Sha, F., Wang, L.:
Characteristics of air pollution episodes influenced by biomass burning pollution in Shanghai, China,
Atmos. Environ, 117756, https://doi.org/10.1016/j.atmosenv.2020.117756, 2020.



Troian-Gautier, L., Turlington, M. D., Wehlin, S. A. M., Maurer, A. B., Brady, M. D., Swords, W. B., and
Meyer, G. J.: Halide photoredox chemistry, Chem. Rev, 119, 4628-4683,
https://doi.org/10.1021/acs.chemrev.8b00732, 2019.

Turnipseed, A. A., Vaghjiani, G. L., Thompson, J. E., and Ravishankara, A. R.: Photodissociation of
$HNO_3$ at 193, 222, and 248 nm: Products and quantum yields, J. Chem. Phys, 96, 5887-5895,
https://doi.org/10.1063/1.462685, 1992.

Vogt, R., Crutzen, P. J., and Sander, R.: A mechanism for halogen release from sea-salt aerosol in the
remote marine boundary layer, Nature, 383, 327, http://dx.doi.org/10.1038/383327a0, 1996.

Wagner, I., Strehlow, H., and Busse, G.: Flash photolysis of nitrate ions in aqueous solution, J. Chem.
Soc. A, 123, 1-33, http://dx.doi.org/10.1524/zpch.1980.123.1.001, 1980.

Wang, G., Zhang, R., Gomez, M. E., Yang, L., Zamora, M. L., Hu, M., Lin, Y., Peng, J., Guo, S., Meng,
691        J., Li, J., Cheng, C., Hu, T., Ren, Y., Wang, Y., Gao, J., Cao, J., An, Z., Zhou, W., Li, G., Wang, J.,
Tian, P., Marrero-Ortiz, W., Secrest, J., Du, Z., Zheng, J., Shang, D., Zeng, L., Shao, M., Wang, W.,
Huang, Y., Wang, Y., Zhu, Y., Li, Y., Hu, J., Pan, B., Cai, L., Cheng, Y., Ji, Y., Zhang, F., Rosenfeld,
D., Liss, P. S., Duce, R. A., Kolb, C. E., Molina. M. J.: Persistent sulfate formation from London
fog to Chinese haze, Proc. Natl. Acad. Sci. USA, 113, 13630–13635,
http://dx.doi.org/10.1073/pnas.1616540113, 2016.

Wingen, L. M., Moskun, A. C., Johnson, S. N., Thomas, J. L., Roeselova, M., Tobias, D. J., Kleinman,
698        M. T., and Finlayson-Pitts, B. J.: Enhanced surface photochemistry in chloride-nitrate ion mixtures,
Phys. Chem. Chem. Phys, 10, 5668-5677, http://dx.doi.org/10.1039/b806613b, 2008.

Xia, S.S., Eugene, A. J., and Guzman, M. I.: Cross photoreaction of glyoxylic and pyruvic acids in model
aqueous aerosol, J. Phys. Chem. A, 122, 6457-6466, http://dx.doi.org/10.1021/acs.jpca.8b05724,
2018.

Xu, W., Kuang, Y., Zhao, C., Tao, J., Zhao, G., Bian, Y., Yang, W., Yu, Y., Shen, C., Liang, L., Zhang, G.,
Lin, W., and Xu, X.: $NH_3$-promoted hydrolysis of $NO_2$ induces explosive growth in HONO, Atmos.
Chem. Phys, 19, 10557-10570, http://dx.doi.org/10.5194/acp-19-10557-2019, 2019.

Yabushita, A., Iida, D., Hama, T., and Kawasaki, M.: Direct observation of OH radicals ejected from
water ice surface in the photoirradiation of nitrate adsorbed on ice at 100 K, J. Phys. Chem. A, 112,
9763-9766, http://dx.doi.org/10.1021/jp804622z, 2008.

Ye, C., Gao, H., Zhang, N., and Zhou, X.: Photolysis of nitric acid and nitrate on natural and artificial
surfaces, Environ. Sci. Technol, 50, 3530-3536, http://dx.doi.org/10.1021/acs.est.5b05032, 2016.

Ye, C., Liu, P., Ma, Z., Xue, C., Zhang, C., Zhang, Y., Liu, J., Liu, C., Sun, X., and Mu, Y.: High $H_2O_2$
concentrations observed during haze periods during the winter in Beijing: Importance of $H_2O_2$
oxidation in sulfate formation, Environ. Sci. Technol. Lett, 5, 757-763,
http://dx.doi.org/10.1021/acs.estlett.8b00579, 2018.

Zhang, K., and Parker, K. M.: Halogen radical oxidants in natural and engineered aquatic systems,
Environ. Sci. Technol, 52, 9579-9594, http://dx.doi.org/10.1021/acs.est.8b02219, 2018.

Zheng, B., Zhang, Q., Zhang, Y., He, K. B., Wang, K., Zheng, G. J., Duan, F. K., Ma, Y. L., and Kimoto,
718        T.: Heterogeneous chemistry: a mechanism missing in current models to explain secondary
inorganic aerosol formation during the January 2013 haze episode in North China, Atmos. Chem.
Phys, 14, 2031-2049, http://dx.doi.org/10.5194/acp-15-2031-2015, 2015a.

Zheng, G., K. Duan, F., Su, H., L. Ma, Y., Cheng, Y., Zheng, B., Zhang, Q., Huang, T., Kimoto, T., Chang,
D., Pöschl, U., Cheng, Y., and B. He, K.: Exploring the severe winter haze in Beijing: The impact
of synoptic weather, regional transport and heterogeneous reactions, Atmos. Chem. Phys, 15, 2969-


2983, http://dx.doi.org/10.5194/acp-15-2969-2015, 2015b.

727                              **Figure Captions**

**Figure 1.** Sulfate formation in the presence and absence of nitrate photolysis and $O_2$
under (a) solar irradiation and (b) 313 nm UV light. The concentrations of $NaHSO_3$ and
the added $NH_4NO_3$ were both 30 mM. The light intensity was 8 mW/cm$^2$.
**Figure 2.** Generation of $NO_x$ during nitrate photolysis under 313 nm UV light.
**Figure 3.** Sulfate formation at different pH values adjusted by the addition of $(NH_4)_2SO_4$
(AS), $NH_4HSO_4$ (ABS) and the mixture of $(NH_4)_2SO_4$ and $NH_4HSO_4$, respectively.
Reaction conditions: 30 mM $NaHSO_3$, 30 mM $NH_4NO_3$, light (8 mW/cm$^2$) and air.
**Figure 4.** Comparison of $NH_4NO_3$ and $NaNO_3$ as the source of $NO_3^-$ for sulfate
formation at different pH. Reaction conditions: 30 mM $NaHSO_3$, 30 mM $NH_4NO_3$ or
$NaNO_3$, light (8 mW/cm$^2$) and air.
**Figure 5.** Inhibition of 2-propanol on aqueous phase oxidation of bisulfite. Reaction
conditions: 30 mM $NaHSO_3$, 30 mM $NH_4NO_3$, 20 mM 2-propanol, light (8 mW/cm$^2$)
and air.
**Figure 6.** $H_2O_2$ produced during the photochemical reaction of nitrate under 313nm
irradiation.
**Figure 7.** (a) Effects of halide ions on aqueous phase oxidation of bisulfite under light
and (b) comparison of the effects of halide ions under dark and light conditions.
Reaction conditions: 30 mM $NaHSO_3$, 30 mM $NH_4NO_3$, 30 mM NaCl, NaBr and NaI,
respectively, light (8 mW/cm$^2$) and air.

747                                  **Tables**
**Table 1.** •OH quantum yields of nitrate photolysis as a function of pH.
**Table 2.** Reactions and their rate constants or quantum yields involved in this study.

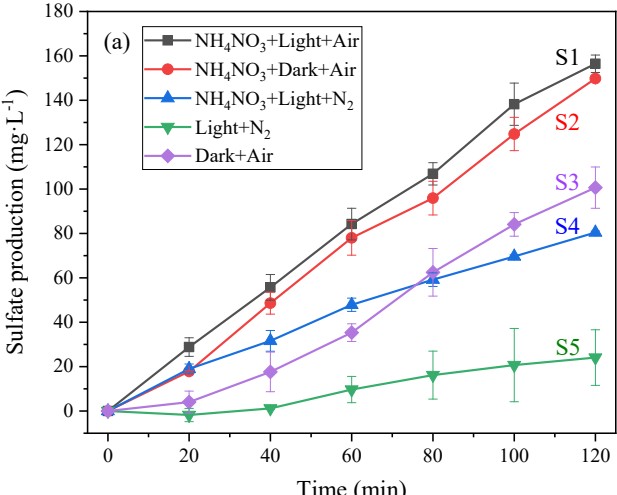


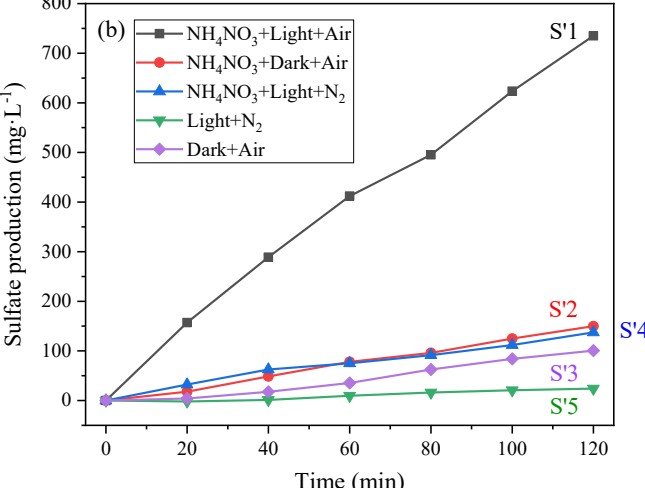


**Figure 1.** Sulfate formation in the presence and absence of nitrate photolysis and $O_2$

under (a) solar irradiation and (b) 313 nm UV light. The concentrations of $NaHSO_3$ and

the added $NH_4NO_3$ were both 30 mM. The light intensity was 8 mW/cm$^2$.



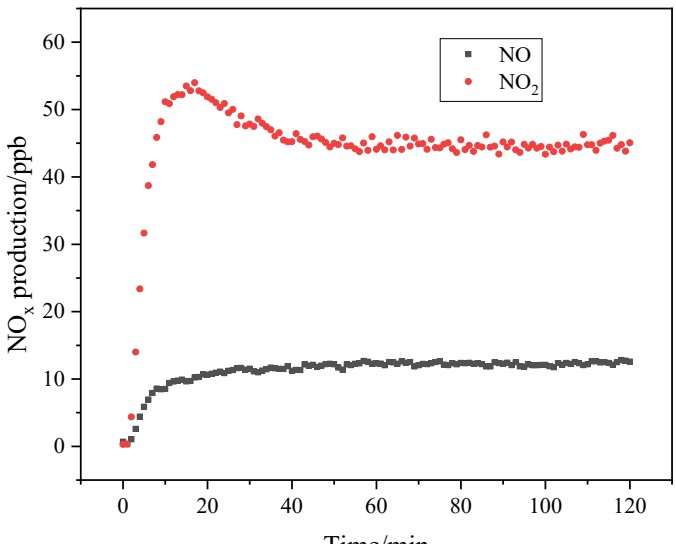


**Figure 2.** Generation of NO$_x$ during nitrate photolysis under 313 nm UV light.

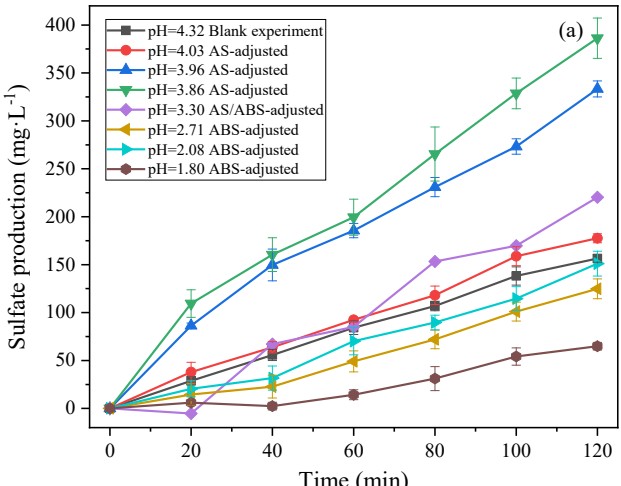


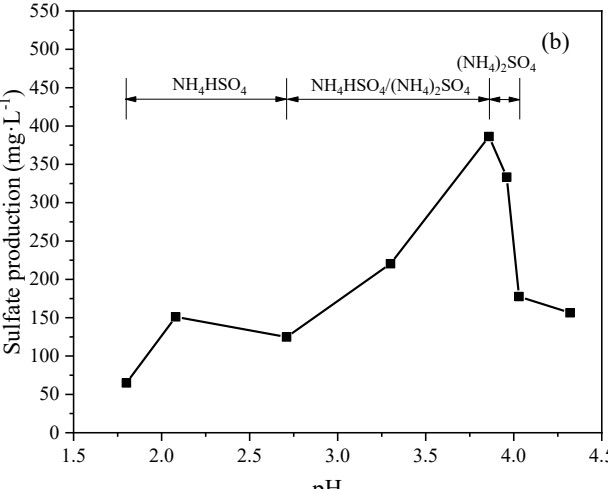


**Figure 3.** Sulfate formation at different pH values adjusted by the addition of $(NH_4)_2SO_4$

(AS), $NH_4HSO_4$ (ABS) and the mixture of $(NH_4)_2SO_4$ and $NH_4HSO_4$, respectively.

Reaction conditions: 30 mM $NaHSO_3$, 30 mM $NH_4NO_3$, light (8 mW/cm$^2$) and air.





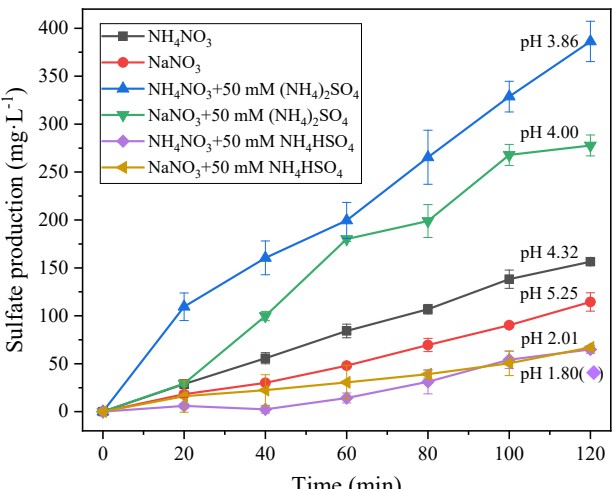


**Figure 4.** Comparison of NH$_4$NO$_3$ and NaNO$_3$ as the source of NO$_3^-$ for sulfate

formation at different pH. Reaction conditions: 30 mM NaHSO$_3$, 30 mM NH$_4$NO$_3$ or

NaNO$_3$, light (8 mW/cm$^2$) and air.













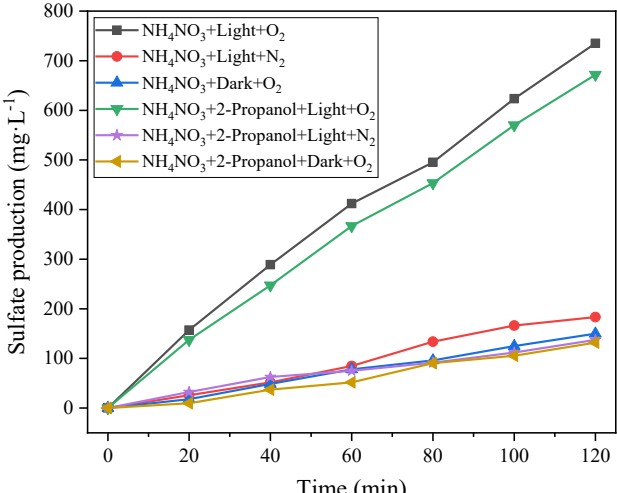


**Figure 5.** Inhibition of 2-propanol on aqueous phase oxidation of bisulfite under 313
nm irradiation. Reaction conditions: 30 mM NaHSO$_3$, 30 mM NH$_4$NO$_3$, 20 mM 2-
propanol, light (8 mW/cm$^2$) and air.





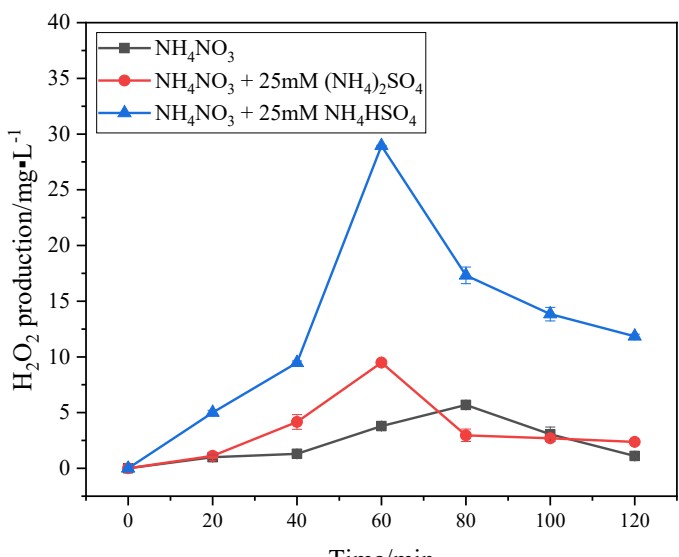

**Figure 6.** $H_2O_2$ produced during the photochemical reaction of nitrate under 313 nm
irradiation.



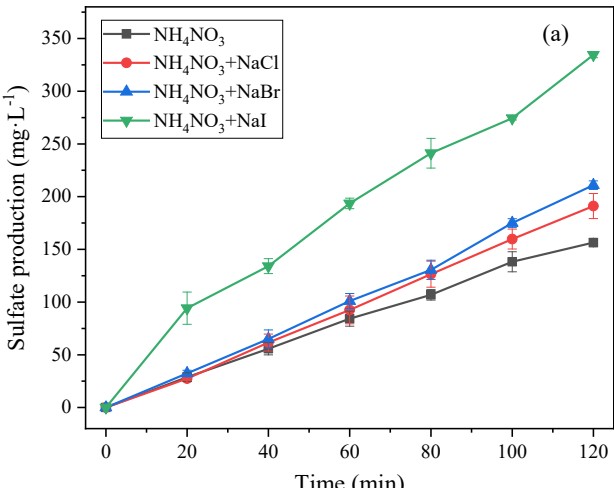

788

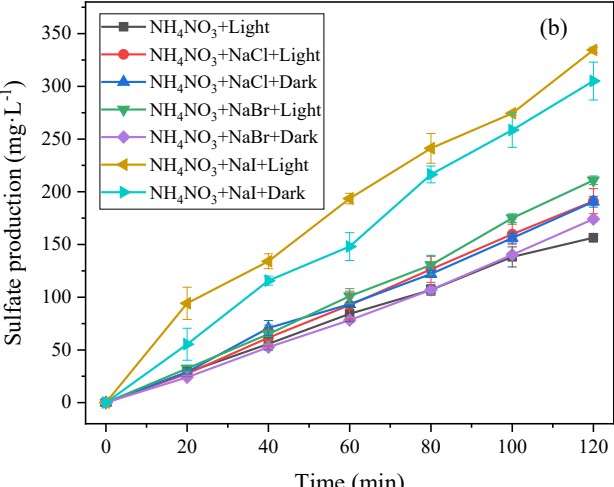

789

**Figure 7.** (a) Effects of halide ions on aqueous phase oxidation of bisulfite under light

and (b) comparison of the effects of halide ions under dark and light conditions.

Reaction conditions: 30 mM NaHSO$_3$, 30 mM NH$_4$NO$_3$, 30 mM NaCl, NaBr and NaI,

respectively, light (8 mW/cm$^2$) and air.












**Table 1.** •OH quantum yields of nitrate photolysis as a function of pH.

| Conditions | pH | $\Phi_{\cdot OH}$ ($\times 10^{-3}$) |
|:---:|:---:|:---:|
| Air | 4.32 | $4.240 \pm 0.353$ |
| Air | 4.03 | $4.033 \pm 0.525$ |
| Air | 3.96 | $7.785 \pm 0.490$ |
| Air | 2.71 | $8.252 \pm 0.226$ |
| Air | 2.08 | $16.672 \pm 0.899$ |






**Table 2.** Reactions and their rate constants or quantum yields involved in this study.

| No. | Reaction | $k$ or $\Phi$ | Ref |
|---|---|---|---|
| R1 | $SO_{2(g)} \overset{k_1}{\Leftrightarrow} SO_2 \cdot H_2O \overset{k_2}{\Leftrightarrow} HSO_3^- + H^+ \overset{k_3}{\Leftrightarrow} SO_3^{2-} + 2H^+$ | $k_1=1.23$; $k_2=1.3\times10^{-2}$; $k_3=6.6\times10^{-8}$ | (Alexander et al., 2003;Gen et al., 2019b) |
| R2 | $HSO_3^- + H_2O_2 \rightarrow HSO_4^- + H_2O$ | $k = 7.45\times10^7$ M$^{-1}$ s$^{-1}$ | (Seinfeld and Pandis, 2006;Ye et al., 2018;Shen et al., 2012) |
| R3 | $NO_3^- + h\nu\,(+\,H^+) \rightarrow NO_2 + \cdot OH$ | $k=8.5\times10^{-7}$ s$^{-1}$; $\Phi$ ($\lambda$=305 nm) = (9.2±0.4) × 10$^{-3}$ | (Scharko et al., 2014;Yabushita et al., 2008) |
| R4 | $NO_3^- + h\nu \rightarrow NO_2 + O^-$ | $\Phi$ ($\lambda > 290$ nm) = 0.01 | (Scharko et al., 2014) |
| R5 | $NO_3^- + h\nu \rightarrow NO_2 + O\,(^3P)$ | $k=8.5\times10^{-8}$ s$^{-1}$; $\Phi$ ($\lambda$=305 nm) = 0.001 | (Scharko et al., 2014) |
| R6 | $2NO_2 + H_2O \rightarrow HONO + H^+ + NO_3^-$ | $k = 1\times10^8$ | (Li et al., 2018b;Richards-Henderson et al., 2013) |
| R7 | $2NO_2 + H_2O + h\nu \rightarrow HONO + \cdot OH$ | $k = 1.7\times10^{-10}$ cm$^3$ mol$^{-1}$ s$^{-1}$ | (Bao et al., 2018;Gligorovski et al., 2015) |
| R8 | $NO_2^- + H_2O \rightarrow HONO + OH^-$ | — | (Yabushita et al., 2008) |
| R9 | $O^- + H^+ / H_3O^+ \rightarrow \cdot OH + H_2O$ | — | (Yabushita et al., 2008) |
| R10 | $HONO(\text{or } NO_2^- + H^+) + h\nu \rightarrow \cdot OH + \cdot NO$ | $k = 1.0\times10^{-3}$ s$^{-1}$; $\Phi$ ($\lambda< 390$ nm ) = 1.0 | (Li et al., 2018b;Scharko et al., 2014) |
| R11 | $NO_2 + HSO_3^- + H_2O \rightarrow 3H^+ + 2NO_2^- + SO_4^{2-}$ | — | (Clifton et al., 1988) |
| R12 | $NO_2 + HSO_3^- + H_2O + O_2 \rightarrow H^+ + NO_3^- + \cdot OH + HSO_4^-$ | — | (Li et al., 2018b) |
| R13 | $NO_2 + h\nu \rightarrow NO + O$ | $k = 1.1\times10^{-2}$ s$^{-1}$ | (Scharko et al., 2014) |
| R14 | $HONO + h\nu \rightarrow NO + OH$ | $k = 1.0\times10^{-3}$ s$^{-1}$ | (Scharko et al., 2014) |
| R15 | $\cdot OH + H_2O_2 \rightarrow H_2O + \cdot HO_2$ | $k = 2.7\times10^7$ M$^{-1}$ s$^{-1}$ | (Hislop and Bolton, 1999;Gligorovski et al., 2015) |
| R16 | $O\,(^3P) + O_2 \rightarrow O_3$ | $k = 4\times10^9$ M$^{-1}$ s$^{-1}$ | (Dubowski et al., 2001;Herrmann, 2007) |
| R17 | $X^- + h\nu \rightarrow X^\cdot + e^-$ | — | (Zhang and Parker, 2018) |
| R18 | $X^\cdot + X^-$ (or $Y^-$) $\rightleftharpoons X_2^{\cdot-}$ (or $XY^{\cdot-}$) | $k =8.5\times10^9$(Cl)/1.6-2.8×10$^9$(Br)/0.1-1.2×10$^{10}$(I) M$^{-1}$ s$^{-1}$ | (Zhang and Parker, 2018;Troian-Gautier et al., 2019) |
| R19 | $\cdot OH + X^- \rightleftharpoons HOX^{\cdot-} \rightleftharpoons OH^- + X^\cdot$ | — | (Zhang and Parker, 2018) |



















**Table 2.** Reactions and their rate constants or quantum yields involved in this study
(continued).

| No. | Reaction | $k$ or $\Phi$ | Ref |
|---|---|---|---|
| R20 | $HOX^{\cdot-} + H^+ \rightleftharpoons H_2O + X^\cdot$ | — | (Zhang and Parker, 2018) |
| R21 | $HOX^{\cdot-} + X^-$ (or $Y^-$) $\rightleftharpoons OH^- + X_2^{\cdot-}$ (or $XY^{\cdot-}$) | — | (Zhang and Parker, 2018) |
| R22 | $SO_4^{\cdot-} + X^- \rightarrow SO_4^{2-} + X^\cdot$ | — | (Zhang and Parker, 2018) |
| R23 | $HSO_5^- + Br^- \rightarrow SO_4^{2-} + HOBr$ | — | (Mozurkewich, 1995) |
| R24 | $HOX + h\nu \rightarrow X^\cdot + {}^\cdot OH$ | — | (Zhang and Parker, 2018) |
| R25 | $X^\cdot + X^\cdot$ (or $Y^\cdot$) $\rightarrow X_2$ (or $XY$) | — | (Zhang and Parker, 2018) |
| R26 | $X^\cdot + X_2^{\cdot-} \rightarrow X_2 + X^-$ | — | (Zhang and Parker, 2018) |
| R27 | $X_2^{\cdot-} + X_2^{\cdot-} \rightarrow X_3^- + X^-$ (or $X_2 + 2X^-$) | $k = 1.9\text{-}9 \times 10^9(Cl)/0.9\text{-}1.2 \times 10^{10}(Br)/3.2\text{-}3.9 \times 10^{10}(I)\ M^{-1}\ s^{-1}$ | (Zhang and Parker, 2018;Troian-Gautier et al., 2019) |
| R28 | $X^\cdot + {}^\cdot OH \rightarrow HOX$ | — | (Zhang and Parker, 2018) |
| R29 | $X_2^{\cdot-} + {}^\cdot OH \rightarrow HOX + X^-$ | — | (Zhang and Parker, 2018) |
| R30 | $HOBr + Br^- + H^+ \rightarrow Br_2 + H_2O$ | $k = 1.6 \times 10^{10}\ L\ mol^{-1}\ s^{-1}$ | (Richards-Henderson et al., 2013;Mozurkewich, 1995) |
| R31 | $HOBr + Cl^- + H^+ \rightarrow BrCl + H_2O$ | $2.3 \times 10^{10}\ M^{-2}\ s^{-2}$ | (Richards and Finlayson-Pitts, 2012;Vogt et al., 1996) |
| R32 | $2^\cdot OH + 2Cl^- \rightarrow Cl_2 + 2OH^-$ | — | (Alexander et al., 2003) |
| R33 | $2^\cdot OH + 2Br^- \rightarrow Br_2 + 2OH^-$ | — | (Richards-Henderson et al., 2013) |
| R34 | $2^\cdot OH + Br^- + Cl^- \rightarrow BrCl + 2OH^-$ | — | (Richards and Finlayson-Pitts, 2012) |
| R35 | $NO_3^- + 3H^+ + 2Br^- \rightarrow Br_2 + HONO + H_2O$ | — | (Richards-Henderson et al., 2013) |
| R36 | $HSO_3^- + HOX \rightarrow 2H^+ + X^- + SO_4^{2-}$ | — | this work |
| R37 | $HSO_3^- + X_2 + H_2O \rightarrow 3H^+ + 2X^- + SO_4^{2-}$ | — | this work |
| R38 | $HSO_3^- + HOCl \rightarrow SO_4^- + 2H^+ + Cl^-$ | — | (Vogt et al., 1996) |
| R39 | $HSO_3^- + HOBr \rightarrow SO_4^- + 2H^+ + Br^-$ | — | (Vogt et al., 1996) |
