# Peer review of "Aqueous phase oxidation of bisulfite influenced by nitrate photolysis"

_Atmospheric Chemistry and Physics, 2020_

## Referee Comment (RC1) · Anonymous Referee #1 · 24 Sep 2020

Chen et al. investigated the formation of sulfate by nitrate photolysis in aqueous solutions. They also examined the effects of the presence of halides on sulfate formation. Nitrate photolysis has recently been proposed as one of the potential missing routes of sulfate formation (Gen et al., 2019ab). The major issue is the novelty of the work and it advances our understanding in light of the available literature. Gen et al. (2019ab) presented results of sulfate formation by nitrate photolysis in droplets upon uptakes of SO2, at 254 and 300 nm respectively. While the experimental approaches are slightly different, these papers essentially work on the same problem as the current ms. In addition, Zhang et al. (2020) also investigated the effect of halide ions on sulfate production rate during nitrate photolysis. They investigated the halide-induced enhancement of nitrate photolysis and potential halogen chemistry on sulfate formation, and

then concluded that halogen chemistry has little effect on sulfate formation compared to halide-induced enhancement of nitrate photolysis in particles. It is a natural expectation that the current ms needs to compare their results with the latest literature. As is the paper presents some interesting results but needs to improve on comparison with the literature results to identify substantiated scientific conclusions and key areas of discrepancies for future research. Furthermore, many discussions are somewhat qualitative, especially in the comparison of experiments done by themselves and others. More quantitative descriptions and comparison of the experimental conditions to go with the result discussions are needed. The interpretation of results without detailed examination of differences in experimental conditions could be erroneous.

1. Line 148: what are the light intensity and wavelength range of the Xe lamp? How is it compared to the 313-nm lamp? Discussions of experimental results should include the light source comparison.

2. Line 158 – 163: Again, please clarify the wavelength range of Xe lamp. ONOO- is one of the important photoproducts at wavelength below 290-nm. If wavelength range of Xe lamp falls in the longer wavelength, OH radical produced via HOONO decomposition may not be important. For example, Goldstein and Rabani observed no ONOO- formation during 300nm illumination ($\varphi$ (ONOO-) < 0.2%) and Benedict et al. suggested that no ONOO- was observed at environmentally relevant wavelengths. In addition, it should be noted that ONOO- can undergo a rapid isomerization to nitrate at pH<6. As a result, ONOO- may not be an important product since pH is below 6 in the current study.

3. Line 177: Can the authors do more to identify which species is most important in the sulfate formation? I would suggest that they conduct a kinetic analysis to identify the contributions of different pathways to sulfate production.

4. Line 186: The authors concluded that nitrate itself can directly oxidize bisulfite by comparing S2 with S3. Kong et al. proposed that nitrate can oxidize sulfate on

hematite, but it does not mean that nitrate can do the same aqueous phase.

5. Line 195 – 198: Zheng et al. (2020) recently incorporated nitrate photolysis pathway into WRF-CMAQ, they found that nitrate photolysis pathway can explain about 15% (assuming an enhancement factor (EF) of 10) to 65% (assuming EF = 100) of the gaps between model estimations and observation in sulfate concentration during winter haze in Beijing. It is one of the very few papers which explicitly examines the role of nitrate photolysis in haze formation and should be cited. The authors in the current ms emphasize quite a bit that the results are consistent with others, including their own previous work, in a qualitative manner. With all these previous works including those already in the references of the current ms, this work needs to attempt to provide more quantitative analysis.

6. Line 200 - 211: Addition of $(NH4)2SO4$ or $NH4HSO4$ can adjust the pH but it can also affect other properties such as ionic strength. How would the authors confirm that other factors are not important? Gen et al. (2019) also investigated the effect of pH on $SO2$ uptake coefficient. They found that the $SO2$ uptake coefficient is not sensitive to initial pH and they attributed it to the similar stable final pH even with different initial pH values. Have the authors measured solution pH as the reaction took place? Variation of pH during the reaction may also affect sulfate production rate. Authors should also investigate a slightly higher pH too (e.g., 5 or 6). Such pH falls in the typical pH range during the haze events in China.

7. Line 246 – 249: Please refer to Figure 4a and 4b in Gen et al. (2019), their results also suggested that sulfate formation rate during $NH4NO3$ photolysis is slightly higher than that during $NaNO3$ photolysis. The type of cation has little influence on the quantum yield of nitrate photolysis, but it does not necessarily mean that the sulfate yield is comparable regardless of type of cations. As authors suggested in the main text, $NH4+$ may play a role in the sulfate formation.

8. Line 263 – 265: Low pH is favorable to the formation of HONO, but the presence

of NH3 will increase pH. Any suggestion why the presence of NH3 can promote the hydrolysis of NO2 and induce the explosive growth of HONO?

9. Line 287 – 289: Have the authors conducted experiments in the absence of NH4NO3, i.e., compared Dark+O2 with 2-propanol+ Dark+O2, which is a direct way to investigate the role of O2 in sulfate formation.

10. Line 291 – 293: It is not clear what the authors would like to prove. The role of O2 in the direct oxidation of HSO3- by nitrate? Or just want to show the direct oxidation of HSO3- by O2? Please clarify the role of O2 in the presence and absence of irradiation.

11. Line 297 - 303: How is O2 related to the OH formation from nitrate photolysis? Gen et al. (2019) did not present that OH generation from nitrate photolysis requires O2. The statement presented in Gen et al. is that oxidation of dissolved SO2 by OH radicals requires O2. Again, it would be useful to compare the contribution of NO2 pathway, NO2- pathway and OH pathway to sulfate formation? Gen et al. (2019) found that direct oxidation of dissolved SO2 by NO2- is an efficient pathway for sulfate formation.

12. Line 309 – 314: In the presence of O2, O(3P) will react with O2 to produce O3. As displayed in the Figure 1a, a larger difference between S1 (NH4NO3+Light+Air) and S4 (NH4NO3+Light+N2) may be also attributed to the O3 pathway.

13. Line 329: Figure S2 is the "First-order photodegradation of 2NB under 313 nm UV irradiation". The trend of H2O2 concentration should be in Figure 6. Please revise.

14. Line 331 – 334: It should be noted that reaction of O3 with OH can produce HO2. Hence, there is no strong evidence to conclude that H2O2 is owing to the recombination of OH.

15. Line 352 – 353: Figure S5 shows that the sulfate production rate under 100 mW/cm2 is comparable to that in 50 mW/cm2. Any suggestions?

16. Line 355: Zhang et al. (2020) found that sulfate production rate will be enhanced

during nitrate photolysis in the presence of halide ions. And they suggested that presence of halide ions can enhance nitrate photolysis, as a result, more oxidants will be produced from nitrate photolysis, which will promote sulfate formation further. While the halide related reactions are possibilities, have the authors conducted any simulation to investigate if enhanced sulfate production is attributed to the enhanced nitrate photolysis? In Zhang et al. (2020) halogen chemistry was included in their box modeling and was not found to play an important role in the enhanced sulfate production.

Reference: Gen, Masao, et al. "Heterogeneous SO2 oxidation in sulfate formation by photolysis of particulate nitrate." Environmental science & technology letters 6.2 (2019a): 86-91.

Gen, Masao, et al. "Heterogeneous Oxidation of SO2 in Sulfate Production during Nitrate Photolysis at 300 nm: Effect of pH, Relative Humidity, Irradiation Intensity, and the Presence of Organic Compounds." Environmental science & technology 53.15 (2019b): 8757-8766.

Goldstein, S.; Rabani, J. Mechanism of Nitrite Formation by Nitrate Photolysis in Aqueous Solutions: The Role of Peroxynitrite, Nitrogen Dioxide, and Hydroxyl Radical. J. Am. Chem. Soc. 2007, 129 (34), 10597−10601.

Benedict, Katherine B., Alexander S. McFall, and Cort Anastasio. "Quantum yield of nitrite from the photolysis of aqueous nitrate above 300 nm." Environmental Science & Technology 51.8 (2017): 4387-4395.

Zheng, Haotian, et al. "Contribution of Particulate Nitrate Photolysis to Heterogeneous Sulfate Formation for Winter Haze in China." Environmental Science & Technology Letters (2020).

Zhang, Ruifeng, et al. "Enhanced Sulfate Production by Nitrate Photolysis in the Presence of Halide Ions in Atmospheric Particles." Environmental Science & Technology 54.7 (2020): 3831-3839.

---

## Referee Comment (RC2) · Anonymous Referee #2 · 23 Oct 2020

The study by Chen et al. explored the influence of nitrate photolysis on sulfate formation from aqueous-phase bisulfite oxidation based on chamber experiments. The role of pH, halogen chemistry and $O_2$ are investigated. This is an interesting topic, and the experiment looks carefully conducted and calibrated. However, the results are poorly organized and reported, with ambiguous data interpretations and conclusions. I don't think the current manuscript meet the standard of ACP, unless it was largely rewritten. Moreover, the authors should treat publications more seriously. Typos are common, but it was very rare to be present at the first line of the abstract—where I believe the "exit" should be "exist". I wonder whether the authors have carefully read the manuscript even once. If even the authors don't want to read the manuscript, neither do the readers.

Major concerns:

1. The study titled "Aqueous phase oxidation of bisulfite influenced by nitrate photolysis", and one would expect the study to focus on reactions under light. In this context, the dark condition is expected to serve as a background. However, this study spends quite a lot of efforts emphasizing the importance of nitrate and its synergism with halogen chemistry in sulfate formations even under dark condition. Moreover, the dark condition is apparently not set as the background scenario, as few comparisons are conducted between dark and with light conditions when the other factors are the same. Either the title or the organization and analysis of this study should be modified.

2. Line 47-49 and 233-239: The review of aerosol pH is biased. The same research group of Guo et al. (2017) have reported much higher pH levels (even higher than 7) in Beijing in some later studies that they participated (e.g., Shi et al., 2017). In addition, some recent studies have further revealed the driving factors of aerosol pH (e.g., Pye et al., 2020; Zheng et al., 2020), which can explain the difference in reported pH levels. These advances should be included.

3. L147-150: The statements are confusing. Which one is insignificant?

4. Line 167-169: What's the point / conclusion of the whole part? Is $NO_2$ an important oxidant or not, based on your experiment results?

5. Line 176-178: The whole paragraph is repeating existing explanations of potential pathways. But what's the relationship with this study? Do the results support / disagree with any of the pathways? If not, this part should be simplified into one to two sentences. Discussion of existing studies that is not related to your results should not be part of a research article.

6. Line 187-188: if "nitrate itself can greatly promote the oxidation of bisulfite in the solution under dark condition" already, how important is the photolysis of nitrate? And what's their relative importance? More quantitative analysis should be conducted, as seems all required experiments are already conducted.

7. Line 213: the pH ranges that are achieved by addition of $(NH_4)_2SO_4$, $NH_4HSO_4$ and their mixtures should be stated here.

8. Line 9-10, 216-219, and 463-465: The behavior of pH in bulk solutions should not be mixed with that of aerosols. While in solutions, the pH can be sensitive to ammonia sulfate concentrations and the relative fractions of ammonia sulfate / ammonia bisulfate added, it was largely controlled by other factors for aerosols (Pye et al., 2020; Zheng et al., 2020).

The results should only be used to infer the dependence of reaction rate on pH, not to be interpreted as the dependence of aerosol pH on ammonia sulfates.

9. Line 223-224: within which pH range?
10. Line 225-227: This possible explanation could be checked with simple calculations. Just scale the formation rates to the pH dependence of bisulfite concentrations.
11. Line 243: What's the new aqueous phase oxidation pathways? And what's the relationship of the sunlight discussions with this section (i.e., the effect of pH on sulfate formation)?
12. Line 246-270: This part seems to argue that cation profile of nitrate is important due to two reasons: (1) $NH_4NO_3$ are with lower pH than $NaNO_3$ at the same concentrations, and (2) $NH_4^+$ can be oxidized by $NO_2$ to help generate ·OH. For the first explanation, as stated above, the influence of chemical compositions on solution pH should not be equaled to that on aerosol pH. Therefore, if the authors want to prove this explanation, the formation rates should be compared at the same pH, not the same concentrations. For the second explanation, not all spontaneous reactions can happen—the reaction rates must also be considered. Are there any experiments / references supporting that these reactions can happen at a reasonable rate under ambient conditions?
13. Line 273-284: What's the point of the long discussions of this part? Proving that 2-propanol can serve as •OH scavenger?
14. Line 294-297: If "$O_2$ has little effect on the generation of •OH by nitrate photolysis", how is it important in sulfate formation with nitrate photolysis? Any explanation of the potential pathways?
15. Line 317-326: Rewrite this paragraph to make the points clearer.
16. Line 336-342: Is the higher $H_2O_2$ under lower pH totally due to the higher •OH productions under lower pH, or is it due to that $H_2O_2$ formation is also favored under low pH even assuming same •OH concentrations?
17. Line 351: how high is the "higher" light intensity? And how sensitive is the formation rate on light intensities?
18. Line 367-391: which steps are proposed in this study? Judging from the manuscript, it seems like all the reactions are already proposed by others, and the study here is just trying to combine them in different ways. If not, clearly state the new pathways/steps proposed in this study and provide the experimental evidences.
19. Line 480-485: This implication part seems abrupt and over-interpreted.

Minor concerns:

There's a lot of grammar errors in this study. A thorough langrage editing is suggested. Some examples follow.

1. Line 1, "exit" should be "exist"
2. Line 154: ":" into ".".
3. Line 145-155: Missing reference of this statement.
4. Line 201: "then" should be "thus".
5. Line 219, 295, etc.: "on the one hand" should be "on one hand". In addition, after "on one hand" there should always be an "on the other hand", which is not seen in, e.g., line 295.
6. Line 221: "may because" is wrong in grammar.

References

Pye, H. O. T., Nenes, A., Alexander, B., Ault, A. P., Barth, M. C., Clegg, S. L., Collett Jr, J. L., Fahey, K. M., Hennigan, C. J., Herrmann, H., Kanakidou, M., Kelly, J. T., Ku, I. T., McNeill, V. F., Riemer, N., Schaefer, T., Shi, G., Tilgner, A., Walker, J. T., Wang, T., Weber, R., Xing, J., Zaveri, R. A., and Zuend, A.: The acidity of atmospheric particles and clouds, Atmos. Chem. Phys., 20, 4809-4888, 10.5194/acp-20-4809-2020, 2020.
Shi, G., Xu, J., Peng, X., Xiao, Z., Chen, K., Tian, Y., Guan, X., Feng, Y., Yu, H., Nenes, A., and Russell, A. G.: pH of Aerosols in a Polluted Atmosphere: Source Contributions to Highly Acidic Aerosol, Environmental Science & Technology, 51, 4289-4296, 10.1021/acs.est.6b05736, 2017.
Zheng, G., Su, H., Wang, S., Andreae, M. O., Pöschl, U., and Cheng, Y.: Multiphase buffer theory explains contrasts in atmospheric aerosol acidity, Science, 369, 1374-1377, 10.1126/science.aba3719, 2020.

---

## Author Comment (AC1) · 6 Dec 2020

Dear Editor and anonymous reviewer,

Thank you very much for your comments on our manuscript [acp-2020-806]. Your comments and suggestions are valuable and helpful for improving our manuscript. We have attempted to address all these comments and given a point-by-point response below. We copy the comments and respond as below. We appreciate for your warm work earnestly, and hope that the revisions and answers are satisfactory. Thank you once again for your time and consideration.

Yours sincerely,

Lingdong Kong

[Figure]

Review 1: ———— Q: Chen et al. investigated the formation of sulfate by nitrate photolysis in aqueous solutions. They also examined the effects of the presence of halides on sulfate formation. Nitrate photolysis has recently been proposed as one of the potential missing routes of sulfate formation (Gen et al., 2019ab). The major issue is the novelty of the work and it advances our understanding in light of the available literature. Gen et al. (2019ab) presented results of sulfate formation by nitrate photolysis in droplets upon uptakes of SO2, at 254 and 300 nm respectively. While the experimental approaches are slightly different, these papers essentially work on the same problem as the current ms. In addition, Zhang et al. (2020) also investigated the effect of halide ions on sulfate production rate during nitrate photolysis. They investigated the halide-induced enhancement of nitrate photolysis and potential halogen chemistry on sulfate formation, and then concluded that halogen chemistry has little effect on sulfate formation compared to halide-induced enhancement of nitrate photolysis in particles. It is a natural expectation that the current ms needs to compare their results with the latest literature. As is the paper presents some interesting results but needs to improve on comparison with the literature results to identify substantiated scientific conclusions and key areas of discrepancies for future research. Furthermore, many discussions are somewhat qualitative, especially in the comparison of experiments done by themselves and others. More quantitative descriptions and comparison of the experimental conditions to go with the result discussions are needed. The interpretation of results without detailed examination of differences in experimental conditions could be erroneous. ————A: As you said, we studied the same topics as some of the latest publications. It is very challenging for us. But we do our best to highlight our novelty and difference. First, the experimental methods were different. We focused on the aqueous-phase oxidation of S(IV). We carried out heterogeneous aqueous-phase reaction of bisulfite under simulated sunlight and 313 nm UV lamp respectively. Secondly, in view of the formation pathway of bisulfite, we found some novel phenomena, such as the promotion of nitrate itself on sulfate formation in the dark, and the generation of the H2O2 by the recombination of OH produced by nitrate photolysis. Thirdly, we tried our best

to conduct more quantitative descriptions, including the contribution of several reaction pathways to sulfate formation. The role of OH in our study is much higher than that in previous reports. Moreover, when studying the effect of pH on sulfate formation by adding ammonium sulfate and ammonium bisulfate, we found and proposed the existence of optimal pH value and the important role of NH4+ in sulfate formation. The explanations and modifications for specific comments are as follows.

1. Line 148: what are the light intensity and wavelength range of the Xe lamp? How is it compared to the 313-nm lamp? Discussions of experimental results should include the light source comparison. ————A: The irradiation of xenon lamp was used to simulate sunlight, which contains a small fraction (<5%) of UV light, and the light intensity and wavelength range of the Xe lamp (CEL-TCX250) we used are shown in Fig. 1. In this study, the use of 313-nm lamp is to better present the reaction phenomena and thus explore the reaction mechanism under specific wavelength, especially the reaction phenomenon under UV light. Meanwhile, the wavelength around 310 nm (e.g. 310 nm, 311 nm, 313 nm) is also the most commonly used wavelength by most studies on nitrate photolysis. For example, Roca et al., J. Phys. Chem. A 2008, 112, 13275–13281(310 nm); Richards, et al., J. Phys. Chem. A 2011, 115, 5810–5821 (311 nm); McFall et al., Environ. Sci. Technol. 2018, 52, 5710−5717 (313 nm). The intensity of the two lamps were both 8 mW/cm2. Corresponding revisions have been made in the revised manuscript. The related comparisons of experimental results are discussed in the section "Aqueous Oxidation of Bisulfite by Nitrate Photolysis". And specific experimental condition of each experiment is also supplemented in Figure captions.

2. Line 158-163: Again, please clarify the wavelength range of Xe lamp. ONOO- is one of the important photoproducts at wavelength below 290-nm. If wavelength range of Xe lamp falls in the longer wavelength, OH radical produced via HOONO decomposition may not be important. For example, Goldstein and Rabani observed no ONOO- formation during 300nm illumination (' (ONOO-) < 0.2%) and Benedict et al. suggested that no ONOO- was observed at environmentally relevant wavelengths. In

addition, it should be noted that ONOO- can undergo a rapid isomerization to nitrate at pH<6. As a result, ONOO- may not be an important product since pH is below 6 in the current study. ————A: Thanks for your precise comments. As mentioned before, the irradiation of xenon lamp contains a small fraction (<5%) of UV light, let alone those less than 290 nm. And the pH of current reaction system was indeed less than 6. Therefore, as you said, ONOO- is not an important photoproduct for OH formation and we deleted this content from the manuscript.

3. Line 177: Can the authors do more to identify which species is most important in the sulfate formation? I would suggest that they conduct a kinetic analysis to identify the contributions of different pathways to sulfate production. ————A: According to the sulfate yields under different conditions, it could be calculated that the role that different pathway played on sulfate formation are as follows: nitrate photolysis (79.6%, in which OH (25.0%)), direct O2 oxidation (13.7%) and nitrate itself (6.7%). However, we are sorry for that it is difficult for us to conduct a kinetic analysis to identify the contribution of NO2 and NO2- due to the lack of more parameters. Corresponding revisions have been made in the revised manuscript.

4. Line 186: The authors concluded that nitrate itself can directly oxidize bisulfite by comparing S2 with S3. Kong et al. proposed that nitrate can oxidize sulfate on hematite, but it does not mean that nitrate can do the same aqueous phase. —————A: Thank you for pointing this out. Originally, we only wanted to use this reference to support the novel phenomenon we found in the aqueous phase. According to your suggestion, this quotation is not suitable. We have deleted this reference in the revised manuscript. Corresponding revisions have been made in the revised manuscript: "This result is consistent with our previous study in which nitrate facilitates the heterogeneous conversion of SO2 on humid hematite particles in the dark (Kong et al., 2014). Furthermore, this result also confirms..." has been changed to "This result may confirm our previous finding that high-nitrate haze episodes favor the heterogeneous aqueous oxidation of SO2 and the formation of sulfate (Kong et al., 2018).".

[Figure]

5. Line 195 – 198: Zheng et al. (2020) recently incorporated nitrate photolysis pathway into WRF-CMAQ, they found that nitrate photolysis pathway can explain about 15% (assuming an enhancement factor (EF) of 10) to 65% (assuming EF = 100) of the gaps between model estimations and observation in sulfate concentration during winter haze in Beijing. It is one of the very few papers which explicitly examines the role of nitrate photolysis in haze formation and should be cited. The authors in the current ms emphasize quite a bit that the results are consistent with others, including their own previous work, in a qualitative manner. With all these previous works including those already in the references of the current ms, this work needs to attempt to provide more quantitative analysis. ————A: According to your suggestion, we have cited the paper by Zheng et al. in the manuscript (line 71). As for more quantitative analysis, according to the sulfate yields under different conditions, it could be calculated that the contributions of the different pathways in sulfate formation are as follows: nitrate photolysis (79.6%, in which OH (25.0%)), direct O2 oxidation (13.7%) and nitrate itself (6.7%). These have been added to the revised manuscript. However, as mentioned in the response to specific comment 3, we are sorry for that it is difficult for us to conduct a kinetic analysis to identify the contribution of NO2 and NO2- due to the lack of more parameters.

6. Line 200 - 211: Addition of (NH4)2SO4 or NH4HSO4 can adjust the pH but it can also affect other properties such as ionic strength. How would the authors confirm that other factors are not important? Gen et al. (2019) also investigated the effect of pH on SO2 uptake coefficient. They found that the SO2 uptake coefficient is not sensitive to initial pH and they attributed it to the similar stable final pH even with different initial pH values. Have the authors measured solution pH as the reaction took place? Variation of pH during the reaction may also affect sulfate production rate. Authors should also investigate a slightly higher pH too (e.g., 5 or 6). Such pH falls in the typical pH range during the haze events in China. ————A: (1) We did take other factors into account after the addition of (NH4)2SO4 (AS) or NH4HSO4 (ABS). We calculated the ionic strength of initial AS-adjusted and ABS-adjusted solutions. Through the comparison

in Fig. 2, we can know that there is a positive correlation between ionic strength and sulfate formation rate in AS-adjusted system, while in ABS-adjusted system, the effect of ionic strength seemed insignificant, and the system with high ionic strength does not have high sulfate formation rate. The system (pH=1.80 ABS-adjusted) with the same highest ionic strength as system (pH=3.86 AS-adjusted) have the lowest sulfate formation rate. This result shows that pH is a much more critical factor on the oxidation of bisulfite, that is what we care more about. And more importantly, we attempt to emphasize the key role of the two main components in ambient atmosphere, (NH4)2SO4 or NH4HSO4. In addition to their own participation in the reaction, they also can regulate pH. (2) The pH values listed in the Tables are all the initial pH values of each reaction. In addition, we also measured the pH at the end of each reaction. However, no valuable information can be found from them. Moreover, we studied the effect of pH on the sulfate oxidation while Gen et al. focused on the effect of pH on SO2 uptake coefficient. we think that the two are not comparable. (3) NH4NO3 and NaHSO3 were selected as the two main reactants in our experiment and 4.23 is the maximum pH values of the solution that could be reached under the optimal concentration ratio. Therefore, the experiment with higher pH (5 or 6) could not be performed. Corresponding revisions have been made in the revised manuscript: Line 245, the following content has been added into the revised manuscript. "Furthermore, the addition of (NH4)2SO4 or NH4HSO4 would affect the ionic strength of the solution, and hence the effect of ionic strength on sulfate formation was analyzed based on the ionic strength of initial (NH4)2SO4-adjusted and NH4HSO4-adjusted solutions. As shown in Text S4 (Table S2), it is found that there was a positive correlation between ionic strength and sulfate formation rate in (NH4)2SO4-adjusted system, but in NH4HSO4-adjusted system, the effect of ionic strength seemed insignificant, and the system with high ionic strength did not have high sulfate formation rate. Therefore, this result shows that pH is a more important factor affecting the oxidation of bisulfite when compared with ionic strength."

7. Line 246 – 249: Please refer to Figure 4a and 4b in Gen et al. (2019), their results also suggested that sulfate formation rate during NH4NO3 photolysis is slightly higher

than that during NaNO3 photolysis. The type of cation has little influence on the quantum yield of nitrate photolysis, but it does not necessarily mean that the sulfate yield is comparable regardless of type of cations. As authors suggested in the main text, NH4+ may play a role in the sulfate formation. ————A: We thank the reviewer for the useful comments and consideration. In fact, besides the role of NH4+ in the sulfate formation, we also investigated the effect of the type of cation on the formation of H2O2. Previous studies haven't detected H2O2 product during steady-state irradiation of NO2- and NO3- solutions at $\lambda > 200$ nm, but Wagner et al. found H2O2 formation in flash photolysis of nitrate ions in aqueous solution (Wagner, et al., Zeitschrift für Physikalische Chemie 1980, 123, 1-33). Mack et al. thought that the combination reaction between two •OH radicals produced by nitrate photolysis is highly unlikely due to the very low concentration and short lifetime of •OH, and then they attributed the formation of H2O2 observed by Wagner et al. to the H2O photolysis at $\lambda > 180$ nm (Mack, et al., J. Photochem. Photobio. A. 1999, 128, 1–13). But it is worth noting that most of the nitrates used in their studies are alkali and alkaline earth metal nitrates, such as sodium nitrate solution, potassium nitrate, calcium nitrate solutions, and magnesium nitrate solution. Besides these studies, Yabushita et al. investigated the photolysis of nitrate produced from nitric acid adsorption onto ice surface, and they found that gaseous •OH production should be attributed to the secondary photolysis of H2O2 produced on ice surface, implying the formation of H2O2 (Yabushita, et al., J. Phys. Chem. A 2008, 112, 9763–9766). However, up to now the photolysis of ammonium nitrate has still received little attention. Recently, we found that H2O2 is difficult to be detected in the photolysis process of sodium nitrate solution, but it is easy to be observed in the solution containing ammonium nitrate, which means that the photolysis behavior of ammonium nitrate solution is different from that of sodium nitrate solution, and thus implies that they would have different effects on the oxidation of bisulfite under light. The effect of cation type on bisulfite oxidation deserves attention. We are preparing another paper on this new finding.

8. Line 263 – 265: Low pH is favorable to the formation of HONO, but the presence

of NH3 will increase pH. Any suggestion why the presence of NH3 can promote the hydrolysis of NO2 and induce the explosive growth of HONO? ————A: The NH3 that was discussed here came from the hydrolysis of NH4+ and was not directly added. As is known to all, the hydrolysis of NH4+ in the aqueous solution can produce H+ and NH3•H2O. The former enhances the acidity of the solution, while the latter produces NH3. The possible reasons why the presence of NH3 can promote the hydrolysis of NO2 and induce the explosive growth of HONO have been illustrated in the manuscript (line 257-267). The main reasons are: (1) NH3 can promote the shift of chemical equilibrium of the hydrolysis of NO2 (reaction R6) and then enhance the formation of HONO; (2) NH3 can promote the hydrolysis of NO2 and induce the explosive growth of HONO via reaction R6 by reducing the free energy barrier of the reaction and stabilizing the product state (Li et al., 2018a; Xu et al., 2019). Appropriate revisions have been made in the revised manuscript.

9. Line 287 – 289: Have the authors conducted experiments in the absence of NH4NO3, i.e., compared Dark+O2 with 2-propanol+ Dark+O2, which is a direct way to investigate the role of O2 in sulfate formation. ————A: Thank you for your suggestions. We haven't conducted experiments in the absence of NH4NO3. This is because that what we attempted to discussed here is that the key role of O2 for the process of nitrate photolysis that affecting the conversion of HSO3-, rather than the direct role of O2 in sulfate formation. Meanwhile, as can be seen from Fig. 1, the direct oxidation of HSO3- by O2 was also verified according to S'3 (8.74 $\mu$MÂůmin-1).

10. Line 291 – 293: It is not clear what the authors would like to prove. The role of O2 in the direct oxidation of HSO3- by nitrate? Or just want to show the direct oxidation of HSO3- by O2? Please clarify the role of O2 in the presence and absence of irradiation. ————A: Here we attempted to discuss the role of O2 in the oxidation of HSO3- by nitrate photolysis rather than its direct oxidation to HSO3-. We pointed out that O2 oxidation is not the main direct contributor on bisulfite oxidation when nitrate photolysis under 313nm UV irradiation, but it could promote the photodegradation of

nitrate or react with O(3P) to form O3 to oxidize bisulfite. However, under dark conditions, O2 directly oxidizes sodium bisulfite and this pathway is the most important way for sulfate formation. Appropriate revisions have been made in the revised manuscript: "The inconspicuous inhibition showed that O2 oxidation pathway is not the main direct contributor to the aqueous phase oxidation of bisulfite when nitrate photolysis occurs under 313 nm UV light."

11. Line 297 - 303: How is O2 related to the OH formation from nitrate photolysis? Gen et al. (2019) did not present that OH generation from nitrate photolysis requires O2. The statement presented in Gen et al. is that oxidation of dissolved SO2 by OH radicals requires O2. Again, it would be useful to compare the contribution of NO2 pathway, NO2- pathway and OH pathway to sulfate formation? Gen et al. (2019) found that direct oxidation of dissolved SO2 by NO2- is an efficient pathway for sulfate formation. ————A: We have modified the misunderstood sentences. And little relationship between O2 and OH formation from nitrate photolysis was found, which we have already expressed in the manuscript. To avoid ambiguity, we revised the explanations for the key role of O2 on sulfate formation (apart from its direct oxidation). That is, in the presence of O2, O(3P) will react with O2 to produce O3, thus forming a new oxidation pathway. And we calculated that the OH pathway contributes 25.0% to the sulfate formation under 313 nm UV light. We are sorry for that it is difficult for us to conduct a kinetic analysis to identify the contribution of NO2 and NO2- due to the lack of more parameters.

12. Line 309 – 314: In the presence of O2, O(3P) will react with O2 to produce O3. As displayed in the Figure 1a, a larger difference between S1 (NH4NO3+Light+Air) and S4 (NH4NO3+Light+N2) may be also attributed to the O3 pathway. ————A: Thank you for your comments. We added the explanation of ozone pathway to the difference between S1 (NH4NO3+Light+Air) and S4 (NH4NO3+Light+N2) under simulated solar irradiation and the difference between S'1 and S'4 under UV light. And, it was also an appropriate explanation for the key role of O2 in sulfate formation (its indirect role

rather than the direct oxidation of bisulfite).

13. Line 329: Figure S2 is the "First-order photodegradation of 2NB under 313 nm UV irradiation". The trend of H2O2 concentration should be in Figure 6. Please revise. ————A: Thank you for pointing this out. It has been revised in the text.

14. Line 331 – 334: It should be noted that reaction of O3 with OH can produce HO2. Hence, there is no strong evidence to conclude that H2O2 is owing to the recombination of OH. ————A: Firstly, we are sorry for that we have not found the literatures on the reaction of O3 with OH to produce HO2. The only literature we found is that O3 reacts with OH- to form HO2 under alkaline conditions (Gligorovski et al., Chem Rev. 115, 13051-13092, 2015). This reaction does not occur in our system (pHïïjIJ4.32). Secondly, Warneck and Wurzinger reported that the quantum yields at 305 nm for reaction of OH with OH to form H2O2 in aqueous solution was $9.2 \times 10\text{-}3$ M-1s-1 (Warneck, P.; Wurzinger, C. J. Phys. Chem. 1988, 92, 6278), while the aqueous-phase oxidation of bisulfite by O3 is rapid, the rate constant of the oxidation of bisulfite by O3 is $3.7 \times 105$ M-1s-1 (Seinfeld, J. H. and Pandis, S. N.: Atmospheric Chemistry and Physics: From Air Pollution to Climate Change, 2nd Edn., John Wiley & Sons Inc., New York, 2006.). Obviously, if O3 is quickly consumed by bisulfite, it will not react with OH. Thirdly, it is generally believed that HO2 mainly comes from the oxidation of CO or VOCs by OH radicals or O3, as well as photolysis of formaldehyde (Lee et al., Atmospheric Environment, 2000, 34, 3475-3494; Schuttlefield et al., J. Am. Chem. Soc., 2008, 130, 12210-12211.). Finally, in PNAS, Lee et al. recently reported experimental evidence that H2O2 is spontaneously produced from pure water by atomizing bulk water into microdroplets, which does not occur in bulk aqueous solutions (Lee et al., PNAS, 2019, 116, 19294-19298), and they also proposed the formation of H2O2 from the recombination of hydroxyl radicals generated from the interaction of electric field with OH−. Therefore, the formation mechanism of H2O2 from the recombination of OH can be supported by some previous studies.

15. Line 352 – 353: Figure S5 shows that the sulfate production rate under 100

mW/cm2 is comparable to that in 50 mW/cm2. Any suggestions? ————A: The experiments with different light intensity were carried out under xenon lamp (simulated sunlight, containing a small fraction (<5%) of UV light), and 30 mM NH4NO3 solution was used. The experiments only changed the light intensity, other reaction conditions were the same. Considering that bisulfite is oxidized rapidly by the products of nitrate photolysis, such as OH and H2O2, while nitrate photolysis is slow (the reaction rate constant of nitrate photolysis even at 305 nm UV light is 10-7 or 10-8 order of magnitude, see Table 2), this phenomenon reflects that the photolysis of nitrate is a rate-determining step in the whole reaction. That is, when the light intensity is higher than 50 mW/cm2, the rate of sulfate formation is nearly limited by the rate of nitrate photolysis.

16. Line 355: Zhang et al. (2020) found that sulfate production rate will be enhanced during nitrate photolysis in the presence of halide ions. And they suggested that presence of halide ions can enhance nitrate photolysis, as a result, more oxidants will be produced from nitrate photolysis, which will promote sulfate formation further. While the halide related reactions are possibilities, have the authors conducted any simulation to investigate if enhanced sulfate production is attributed to the enhanced nitrate photolysis? In Zhang et al. (2020) halogen chemistry was included in their box modeling and was not found to play an important role in the enhanced sulfate production. ————A: This is a good suggestion. From the point of view of chemical equilibrium shift, the introduction of halogen ions consumes the oxidizing species produced from nitrate photolysis, which will promote the photolysis of nitrate and produce more new oxidizing species. These new species will promote sulfate formation further. In our paper, we give a qualitative explanation of the results, but not a quantitative explanation through modelling. Although the modelling may give some valuable quantitative results, we can't give the calculation results in a short time because we lack experience in model calculation. Therefore, we are sorry that we have not been able to explain it by using the results of the model calculation, and your understanding is very much appreciated.

Review 2: —————Q: The study by Chen et al. explored the influence of nitrate photolysis on sulfate formation from aqueous-phase bisulfite oxidation based on chamber experiments. The role of pH, halogen chemistry and O2 are investigated. This is an interesting topic, and the experiment looks carefully conducted and calibrated. However, the results are poorly organized and reported, with ambiguous data interpretations and conclusions. I don't think the current manuscript meet the standard of ACP, unless it was largely rewritten. Moreover, the authors should treat publications more seriously. Typos are common, but it was very rare to be present at the first line of the abstract—where I believe the "exit" should be "exist". I wonder whether the authors have carefully read the manuscript even once. If even the authors don't want to read the manuscript, neither do the readers. —————A: We are sorry to make so many mistakes. According to your suggestions, we rearranged the structure of the manuscript, carefully proofread the experimental data and their explanations, and provided as many quantitative descriptions as possible to highlight our novelty compared with existing studies. We have seriously corrected the grammatical errors and checked the full text as well. Thanks for your care and patience. The explanations and modifications for specific comments are as follows.

Major concerns: 1. The study titled "Aqueous phase oxidation of bisulfite influenced by nitrate photolysis", and one would expect the study to focus on reactions under light. In this context, the dark condition is expected to serve as a background. However, this study spends quite a lot of efforts emphasizing the importance of nitrate and its synergism with halogen chemistry in sulfate formations even under dark condition. Moreover, the dark condition is apparently not set as the background scenario, as few comparisons are conducted between dark and with light conditions when the other factors are the same. Either the title or the organization and analysis of this study should be modified. —————A: The title and content of the manuscript have been adjusted as a whole, and more comparisons have been made between dark and with light conditions. The title is changed to "Aqueous phase oxidation of bisulfite influenced by nitrate and its photolysis".

2. Line 47-49 and 233-239: The review of aerosol pH is biased. The same research group of Guo et al. (2017) have reported much higher pH levels (even higher than 7) in Beijing in some later studies that they participated (e.g., Shi et al., 2017). In addition, some recent studies have further revealed the driving factors of aerosol pH (e.g., Pye et al., 2020; Zheng et al., 2020), which can explain the difference in reported pH levels. These advances should be included. ————A: According to your suggestions, the literatures mentioned above have been cited, and appropriate revisions have been made in the revised manuscript.

3. L147-150: The statements are confusing. Which one is insignificant? ————A: This refers to the experiments under xenon lamp light source, which has an insignificant sulfate yield. To avoid misunderstanding, the sentence "Considering the insignificant increase of sulfate yield, ..." is changed to "Considering the insignificant increase of sulfate yield under Xe lamp light source, ...".

4. Line 167-169: What's the point / conclusion of the whole part? Is NO2 an important oxidant or not, based on your experiment results? ————A: Gen et al. reported that NO2 is one of the contributors to sulfate formation. Sarwar et al., (Sarwar et al., Atmos. Environ., 2013, 68, 186-197) reported that the aqueous-phase oxidation of S(IV) by NO2 increases mean winter sulfate by 4-20%. Here we wanted to verify this NO2 pathway for sulfate formation in our reaction system through the observation of the formation of NO2 during nitrate photolysis. However, we are sorry for that it is difficult for us to conduct a kinetic analysis or modelling to identify the contribution of the formed NO2 in our aqueous-phase system due to the lack of more experimental data. Appropriate revisions have been made in the revised manuscript: Line 169: "The formation of gaseous NO2 during the photolysis of nitrate may reveal the aqueous-phase oxidation of bisulfite by NO2 pathway." has been added.

5. Line 176-178: The whole paragraph is repeating existing explanations of potential pathways. But what's the relationship with this study? Do the results support / disagree with any of the pathways? If not, this part should be simplified into one to two

sentences. Discussion of existing studies that is not related to your results should not be part of a research article. ————A: Thank you for your suggestions. According to your suggestions, we have made appropriate modifications: Line 173: "as well by previous studies" has been deleted. Line 174-178: the sentence has been deleted. In addition, on the basis of the existing potential pathways, we conducted more experiments and found some novel phenomena, such as the promotion of sulfate formation by nitrate itself in the dark, the generation of the H2O2 by the recombination of OH produced by nitrate photolysis, the formation of O3 from the reaction of O2 and O(3P). And, the role of OH in our study is much higher than that in previous reports. These results are arranged separately in the original manuscript. In order to avoid misunderstanding, we adjust the structure of the article and describe the differences and novelties respectively.

6. Line 187-188: if "nitrate itself can greatly promote the oxidation of bisulfite in the solution under dark condition" already, how important is the photolysis of nitrate? And what's their relative importance? More quantitative analysis should be conducted, as seems all required experiments are already conducted. ————A: Thank you very much for your suggestions. According to the sulfate yield under 313 nm UV light, the roles played by several pathways are as follows: nitrate itself accounts for about 6.7%, oxygen accounts for about 13.7%, and nitrate photolysis accounts for 79.6%. However, under the irradiation of xenon lamp, the weak nitrate photolysis lead to the low sulfate yield. In this situation, the role of nitrate itself in dark is nearly comparable to that of nitrate photolysis (see Fig. 1a). This quantitative analysis result has been added to the manuscript.

7. Line 213: the pH ranges that are achieved by addition of (NH4)2SO4, NH4HSO4 and their mixtures should be stated here. ————A: Thank you for pointing this out. Accepted and modified. Line 213: The pH ranges (1.80~4.32) are added in the revised manuscript.

8. Line 9-10, 216-219, and 463-465: The behavior of pH in bulk solutions should not

be mixed with that of aerosols. While in solutions, the pH can be sensitive to ammonia sulfate concentrations and the relative fractions of ammonia sulfate / ammonia bisulfate added, it was largely controlled by other factors for aerosols (Pye et al., 2020; Zheng et al., 2020). The results should only be used to infer the dependence of reaction rate on pH, not to be interpreted as the dependence of aerosol pH on ammonia sulfates. ————————A: As you mentioned, the atmospheric aerosol pH is affected by many factors, such as RH and the compositions and concentrations of inorganic and organic aerosols, not only by ammonium sulfate/ammonia bisulfate. Atmospheric aerosols include aqueous phase aerosols (e.g. fog and cloud droplets), aerosol particles with water film, and so on. In our study, the aqueous phase oxidation of bisulfite has been investigated. Considering that secondary inorganic ions are one of the most important components in PM2.5, we selected ammonium sulfate/ammonia bisulfate to adjust solution pH. For example, secondary inorganic aerosol (SIA: $SO_4^{2-}$, $NO_3^-$ and $NH_4^+$) accounted for 54.1% of PM2.5 and 91.0% of the total water-soluble inorganic ions (TWSI) during haze periods, respectively. While during the non-haze periods, SIA accounted for 43.5% of PM2.5 and 90.0% of the TWSI, respectively (Kong et al., Sci. Total. Environ. 2018, 634,1192-1204). Also, hygroscopic SIA easily make particles into droplets under high relative humidity. Of course, the role of ammonium sulfate in adjusting the pH of liquid aerosol particles should not be ignored. In addition, we didn't claim that the behavior of pH in bulk solutions represents the aerosol. What we discussed is only one of the possible cases of that in aqueous phase aerosols. Moreover, it is not mentioned in the manuscript that the pH of aerosol is dependent on ammonium sulfate. We only proposed the crucial role of ammonium sulfate in regulating pH of solutions in the enhancement of aqueous phase sulfate formation, which is of great significance for understanding the behavior of ammonium sulfate in the atmosphere. Therefore, in the revised manuscript, we emphasize that ammonium sulfate regulates the pH of the solution rather than all atmospheric aerosols.

9. Line 223-224: within which pH range? ————————A: The pH range here is within 3.86-4.32, which is adjusted by the addition of different content of (NH4)2SO4. Corresponding revision has been made in the revised manuscript: Line 224: "(within the pH range of 3.86-4.32)" is added.

10. Line 225-227: This possible explanation could be checked with simple calculations. Just scale the formation rates to the pH dependence of bisulfite concentrations. ————A: Thank you for your good suggestion. Bisulfite concentrations under different pH were listed as Fig. 3. As shown in the following Table, $HSO_3^-$ concentration is greatly reduced (about 62% of the analytical concentration) when pH < 2.08, and thus the sulfate yield was suppressed (The sulfate yield at pH=2.08 in the table is slightly higher than that at pH=2.72, which is probably due to experimental error).

11. Line 243: What's the new aqueous phase oxidation pathways? And what's the relationship of the sunlight discussions with this section (i.e., the effect of pH on sulfate formation)? ————A: (1) We are sorry about that we used the inappropriate words. What we wanted to express here is that the role of $(NH_4)_2SO_4$ and $NH_4HSO_4$ in regulating the pH, which is significant for nitrate photolysis. (2) In this section, we discussed the effect of pH on sulfate formation by nitrate photolysis. Because our experiments were carried out under acidic conditions, and the ambient atmospheric aerosol particle is usually acidic under severe haze weather as reported. Therefore, the sunlight became an important factor affecting the reaction. However, according to the literature, the photochemical activity is still maintained at a high level in the severe haze weather. Hence, our experiments and conclusion drew here are reasonable. Corresponding revisions have been made in the manuscript.

12. Line 246-270: This part seems to argue that cation profile of nitrate is important due to two reasons: (1) $NH_4NO_3$ are with lower pH than $NaNO_3$ at the same concentrations, and (2) $NH_4^+$ can be oxidized by $NO_2$ to help generate OH. For the first explanation, as stated above, the influence of chemical compositions on solution pH should not be equaled to that on aerosol pH. Therefore, if the authors want to prove this explanation, the formation rates should be compared at the same pH, not the same concentrations. For the second explanation, not all spontaneous reactions

Interactive
comment

can happen—the reaction rates must also be considered. Are there any experiments / references supporting that these reactions can happen at a reasonable rate under ambient conditions? ————A: Thank you for your suggestions. (1) In this part, we found and proposed the important role of cations in the photolysis of nitrate to promote sulfate formation. Therefore, it is necessary to ensure the same initial concentration of $NH_4^+$ and $Na^+$ cations in the solutions rather than the same pH. In this situation, $NH_4^+$ favored a higher sulfate formation for that the hydrolysis of $NH_4^+$ may maintain a stable and low pH of the solution during the process, which is more conducive to the formation of sulfate as described by R3-R10. What we emphasized here is the role of $NH_4^+$ rather than pH. In addition, we didn't claim that the results of the study in solutions are representative of the aerosol. In fact, atmospheric aerosols include aqueous phase aerosols (e.g. fog and cloud droplets), aerosol particles with water film, and so on. What we discussed is only one of the possible cases of that in liquid aerosols. (2) We agree with you. That is, not all spontaneous reactions can happen—the reaction rates must also be considered. There are some reports about the thermal reaction of $NH_3$ with $NO_2$ (not the selective catalytic reduction of nitrogen oxides ($NO_x$) by $NH_3$ ($NH_3$-SCR)), which is kinetically relevant to the deNOx process. The reported reactions are usually carried out at high temperature. In addition, although on the internet there is an experiment about the rapid generation of white smoke by mixing gaseous $NO_2$ and $NH_3$ at room temperature, we found that there are not any formal reports supporting that these reactions can happen at a reasonable rate under ambient conditions. Therefore, we have deleted this content in our manuscript. Corresponding revisions have been made in the manuscript and supporting information.

13. Line 273-284: What's the point of the long discussions of this part? Proving that 2-propanol can serve as *OH scavenger? ————A: Our purpose is to discuss the role of oxygen and OH in the formation of sulfate. 2-propanol is only used as a tool for its inhibition on oxygen and OH. To avoid misunderstanding, this section is integrated into Section "Aqueous Oxidation of Bisulfite by Nitrate Photolysis".

14. Line 294-297: If "O2 has little effect on the generation of OH by nitrate photolysis", how is it important in sulfate formation with nitrate photolysis? Any explanation of the potential pathways? ————A: O2 has little effect on the generation of OH by nitrate photolysis, but has an important effect on the formation of sulfate. It is reasonably inferred that O(3P), one of the products of nitrate photolysis (NO3- + h$\nu$ →NO2- + O(3P)), will react with O2 to produce O3 (O(3P) +O2→ O3), thus becoming an important contributor for sulfate formation.

15. Line 317-326: Rewrite this paragraph to make the points clearer. ————A: We rewrite this paragraph as follow. "Atmospheric H2O2 plays an important role in sulfate formation, and its formation has attracted much attention. Previous studies haven't detected H2O2 formation during steady-state irradiation of NO2- and NO3- solutions at $\lambda$ > 200 nm (Daniels et al., 1968; Shuali et al., 1969; Mark et al., 1996), but Wagner et al. once found H2O2 formation in flash photolysis of nitrate ions in acidic aqueous solution (Wagner et al., 1980). Yabushita et al. once again found H2O2 formation during the photolysis of nitrate originated from nitric acid adsorption under low-temperature ice conditions, and they attributed its formation to the recombination of the •OH produced by nitrate photolysis (Yabushita et al., 2008). However, Mack et al. thought that the combination reaction between two •OH produced by nitrate photolysis is highly unlikely due to the very low concentration and short lifetime of •OH, and they attributed the formation of H2O2 observed by Wagner et al. to the H2O photolysis at $\lambda$ > 180 nm (Mack and Bolton, 1999). The formation of H2O2 during the photolysis of nitrate remains controversial, which is worth exploring further."

16. Line 336-342: Is the higher H2O2 under lower pH totally due to the higher OH productions under lower pH, or is it due to that H2O2 formation is also favored under low pH even assuming same OH concentrations? ————A: As is shown in Table 1, the quantum yield of OH obviously enhanced as pH decreased, illustrating that the lower pH favored the higher OH production. Meanwhile, higher H2O2 formation was also occurred. We can't exclude the role of lower pH on higher H2O2 formation. Roth

et al. once studied the H2O2 formation by the combination of OH in the radiolysis of water. They verified that higher primary H2O2 yields were occurred at a low pH (<4), while the yields decrease at higher pH too. In addition, it is worth pointing out that H2O2 increased first and then decreased during UV irradiation, which may indicate that the secondary photolysis reaction occurs during H2O2 formation (see Fig. 6).

17. Line 351: how high is the "higher" light intensity? And how sensitive is the formation rate on light intensities? ————A: The experiments with different light intensity were carried out under xenon lamp (simulated sunlight, containing a small fraction (<5%) of UV light), and 30 mM NH4NO3 solution was used. The experiments only changed the light intensity, other reaction conditions were the same. Fig. S5 shows the results of the effect of light intensity on aqueous-phase sulfate formation. Firstly, as can be seen from Fig. S5, the sulfate production rate under 100 mW/cm2 is comparable to that in 50 mW/cm2. Considering that bisulfite is oxidized rapidly by the products of nitrate photolysis, such as OH and H2O2, while nitrate photolysis is slow (the reaction rate constant of nitrate photolysis even at 305 nm UV light is 10-7 or 10-8 order of magnitude, see Table 2), this phenomenon reflects that the photolysis of nitrate is a rate-determining step in the whole reaction. That is, when the light intensity is higher than 50 mW/cm2, the rate of sulfate formation is nearly limited by the rate of nitrate photolysis. Secondly, the sulfate formation rate under 8, 50 and 100 mW/cm2 was 13.58, 22.56, 22.36 respectively. We found that the increase of sulfate formation rate is obvious when the light intensity increases from 8 mW/cm2 to 50 mW/cm2.

18. Line 367-391: which steps are proposed in this study? Judging from the manuscript, it seems like all the reactions are already proposed by others, and the study here is just trying to combine them in different ways. If not, clearly state the new pathways/steps proposed in this study and provide the experimental evidences. ————A: As explained before, on the basis of the existing potential pathways, we conduct more experiments and found some novel phenomena. Firstly, in view of the formation pathway of bisulfite, we found the promotion of nitrate itself on sulfate formation in the dark, and the generation of the H2O2 (Fig. 6) by the recombination of OH produced by nitrate photolysis. Thirdly, we tried our best to conduct more quantitative descriptions, including the contribution of several reaction pathways to sulfate formation. For example, under 313 nm UV light, the contribution of OH in our study is much higher (25.0%) than that in reported paper (<1%). Moreover, when studying the effect of pH on sulfate formation by adding ammonium sulfate and ammonium bisulfate, we found and proposed the existence of optimal pH ($\sim$3.86) value and the important role of NH4+ in sulfate formation. And, halide-induced enhancement of sulfate formation by nitrate photolysis owing to halide photochemistry and the redox cycle of halogen was proposed for the first time in this study.

19. Line 480-485: This implication part seems abrupt and over-interpreted. ————A: This part of implication has been modified appropriately.

Minor concerns: There's a lot of grammar errors in this study. A thorough langrage editing is suggested. Some examples follow. 1. Line 1, "exit" should be "exist". 2. Line 145, ", see Text S1 in Supporting information, hereafter" is added. 3. Line 145-155: Missing reference of this statement. 4. Line 169: "...stable rate" is changed to "...stable formation rate". 5. Line 181-182: "oxygen" is changed to "O2". 6. Line 201: "then" should be "thus". 7. Line 219, 295, etc.: "on the one hand" should be "on one hand". In addition, after "on one hand" there should always be an "on the other hand", which is not seen in, e.g., line 295. 8. Line 221: "may because" is wrong in grammar. ————A: Thanks for your careful comments. The mistakes listed above have been modified in the manuscript and we checked the whole contents more seriously as well.

Please also note the supplement to this comment:
https://acp.copernicus.org/preprints/acp-2020-806/acp-2020-806-AC1-supplement.pdf

———————————————————

[Figure]

[Figure]

**Fig. 1.** The spectrum of the Xe lamp coupled with an optical fiber (model CEL-TCX250).

| pH | AS/ABS-adjusted | Ionic strength (M) | Sulfate formation rate (µM·min⁻¹) |
|---|---|---|---|
| 4.32 | —— | 0.06 | 2.72 |
| 4.03 | AS | 0.0725 | 3.08 |
| 3.96 | AS | 0.135 | 5.79 |
| 3.86 | AS | 0.21 | 6.71 |
| 2.71 | ABS | 0.0725 | 2.17 |
| 2.08 | ABS | 0.135 | 2.62 |
| 1.80 | ABS | 0.21 | 1.13 |

**Fig. 2.** Ionic strength of AS-adjusted and ABS-adjusted solutions

| pH | 1.8 | 2.08 | 2.71 | 3.3 | 3.86 | 3.96 | 4.03 | 4.32 |
|---|---|---|---|---|---|---|---|---|
| $c(H^+)$ | 0.0159 | 0.00791 | 0.002 | 0.0005 | 0.000141 | 0.000112 | 0.000891 | 0.00005 |
| A | 0.000459511 | 0.000165399 | 3.00009E-05 | 6.75086E-06 | 1.85374E-06 | 1.4694E-06 | 1.23777E-05 | 6.53358E-07 |
| $\alpha(HSO_3^-)$-Distribution coefficient | 44.98% | 62.17% | 86.66% | 96.28% | 98.88% | 99.09% | 93.58% | 99.49% |
| $HSO_3^-$(mM) | 13.49 | 18.65 | 26.00 | 28.89 | 29.66 | 29.73 | 28.07 | 29.85 |
| $SO_4^{2-}(\mu M \cdot min^{-1})$ sulfate formation rate | 5.63 | 13.12 | 10.83 | 19.13 | 33.53 | 28.94 | 15.41 | 13.58 |

*: The results in the table are calculated according to the sulfate formation rate under xenon lamp

**Fig. 3.** The content of HSO3- under different pH

**Supplement:**

Dear Editor and anonymous reviewer,

Thank you very much for your comments on our manuscript [acp-2020-806]. Your comments and suggestions are valuable and helpful for improving our manuscript. We have attempted to address all these comments and given a point-by-point response below. We copy the comments and respond as below.

We appreciate for your warm work earnestly, and hope that the revisions and answers are satisfactory.

Thank you once again for your time and consideration.

Yours sincerely,

Lingdong Kong

**Review 1:**

Chen et al. investigated the formation of sulfate by nitrate photolysis in aqueous solutions. They also examined the effects of the presence of halides on sulfate formation. Nitrate photolysis has recently been proposed as one of the potential missing routes of sulfate formation (Gen et al., 2019ab). The major issue is the novelty of the work and it advances our understanding in light of the available literature. Gen et al. (2019ab) presented results of sulfate formation by nitrate photolysis in droplets upon uptakes of $SO_2$, at 254 and 300 nm respectively. While the experimental approaches are slightly different, these papers essentially work on the same problem as the current ms. In addition, Zhang et al. (2020) also investigated the effect of halide ions on sulfate production rate during nitrate photolysis. They investigated the halide-induced enhancement of nitrate photolysis and potential halogen chemistry on sulfate formation, and then concluded that halogen chemistry has little effect on sulfate formation compared to halide-induced enhancement of nitrate photolysis in particles. It is a natural expectation that the current ms needs to compare their results with the latest literature. As is the paper presents some interesting results but needs to improve on comparison with the literature results to identify substantiated scientific conclusions and key areas of discrepancies for future research. Furthermore, many discussions are somewhat qualitative, especially in the comparison of experiments done by themselves and others. More quantitative descriptions and comparison of the experimental conditions to go with the result discussions are needed. The interpretation of results without detailed examination of differences in experimental conditions could be erroneous.

As you said, we studied the same topics as some of the latest publications. It is very challenging for us. But we do our best to highlight our novelty and difference. First, the experimental methods were different. We focused on the aqueous-phase oxidation of S(IV). We carried out heterogeneous aqueous-phase reaction of bisulfite under simulated sunlight and 313 nm UV lamp respectively. Secondly, in view of the formation pathway of bisulfite, we found some novel phenomena, such as the promotion of nitrate itself on sulfate formation in the dark, and the generation of the $H_2O_2$ by the recombination of OH produced by nitrate photolysis. Thirdly, we tried our best to conduct more quantitative descriptions, including the contribution of several reaction pathways to sulfate formation. The role of OH in our study is much higher than that in previous reports. Moreover, when studying the effect of pH on sulfate formation by adding ammonium sulfate and ammonium bisulfate, we found and proposed the existence of optimal pH value and the important role of $NH_4^+$ in sulfate formation.

The explanations and modifications for specific comments are as follows.

1. Line 148: what are the light intensity and wavelength range of the Xe lamp? How is it compared to the 313-nm lamp? Discussions of experimental results should include the light

source comparison.

The irradiation of xenon lamp was used to simulate sunlight, which contains a small fraction (<5%) of UV light, and the light intensity and wavelength range of the Xe lamp (CEL-TCX250) we used are shown in Fig. 1.

[Figure]

Figure 1. The spectrum of the Xe lamp coupled with an optical fiber (model CEL-TCX250).

In this study, the use of 313-nm lamp is to better present the reaction phenomena and thus explore the reaction mechanism under specific wavelength, especially the reaction phenomenon under UV light. Meanwhile, the wavelength around 310 nm (e.g. 310 nm, 311 nm, 313 nm) is also the most commonly used wavelength by most studies on nitrate photolysis. For example, Roca et al., J. Phys. Chem. A 2008, 112, 13275–13281(310 nm); Richards, et al., J. Phys. Chem. A 2011, 115, 5810–5821 (311 nm); McFall et al., Environ. Sci. Technol. 2018, 52, 5710−5717 (313 nm).

The intensity of the two lamps were both 8 mW/cm$^2$.

Corresponding revisions have been made in the revised manuscript. The related comparisons of experimental results are discussed in the section "*Aqueous Oxidation of Bisulfite by Nitrate Photolysis*". And specific experimental condition of each experiment is also supplemented in Figure captions.

2. Line 158-163: Again, please clarify the wavelength range of Xe lamp. ONOO$^-$ is one of the important photoproducts at wavelength below 290-nm. If wavelength range of Xe lamp falls in the longer wavelength, OH radical produced via HOONO decomposition may not be important. For example, Goldstein and Rabani observed no ONOO- formation during 300nm illumination (' (ONOO-) < 0.2%) and Benedict et al. suggested that no ONOO- was observed at environmentally relevant wavelengths. In addition, it should be noted that ONOO- can undergo a rapid isomerization to nitrate at pH<6. As a result, ONOO- may not be an important product since pH is below 6 in the current study.

   Thanks for your precise comments. As mentioned before, the irradiation of xenon lamp contains a small fraction (<5%) of UV light, let alone those less than 290 nm. And the pH of current reaction system was indeed less than 6. Therefore, as you said, ONOO$^-$ is not an important photoproduct for OH formation and we deleted this content from the manuscript.

3. Line 177: Can the authors do more to identify which species is most important in the sulfate formation? I would suggest that they conduct a kinetic analysis to identify the contributions of different pathways to sulfate production.

   According to the sulfate yields under different conditions, it could be calculated that the role that different pathway played on sulfate formation are as follows: nitrate photolysis (79.6%, in which OH (25.0%)), direct $O_2$ oxidation (13.7%) and nitrate itself (6.7%). However, we are sorry for that it is difficult for us to conduct a kinetic analysis to identify the contribution of $NO_2$ and $NO_2^-$ due to the lack of more parameters. Corresponding revisions have been made in the revised manuscript.

4. Line 186: The authors concluded that nitrate itself can directly oxidize bisulfite by comparing S2 with S3. Kong et al. proposed that nitrate can oxidize sulfate on hematite, but it does not mean that nitrate can do the same aqueous phase.

   Thank you for pointing this out.

   Originally, we only wanted to use this reference to support the novel phenomenon we found in the aqueous phase. According to your suggestion, this quotation is not suitable. We have deleted this reference in the revised manuscript.

   Corresponding revisions have been made in the revised manuscript:

   "This result is consistent with our previous study in which nitrate facilitates the heterogeneous conversion of $SO_2$ on humid hematite particles in the dark (Kong et al., 2014). Furthermore, this result also confirms…" has been changed to "This result may confirm our previous finding that high-nitrate haze episodes favor the heterogeneous aqueous oxidation of $SO_2$ and the formation of sulfate (Kong et al., 2018).".

5. Line 195 – 198: Zheng et al. (2020) recently incorporated nitrate photolysis pathway into WRF-CMAQ, they found that nitrate photolysis pathway can explain about 15% (assuming an enhancement factor (EF) of 10) to 65% (assuming EF = 100) of the gaps between model estimations and observation in sulfate concentration during winter haze in Beijing. It is one of the very few papers which explicitly examines the role of nitrate photolysis in haze formation and should be cited. The authors in the current ms emphasize quite a bit that the results are consistent with others, including their own previous work, in a qualitative manner. With all these previous works including those already in the references of the current ms, this work needs to attempt to provide more quantitative analysis.

According to your suggestion, we have cited the paper by Zheng et al. in the manuscript (line 71). As for more quantitative analysis, according to the sulfate yields under different conditions, it could be calculated that the contributions of the different pathways in sulfate formation are as follows: nitrate photolysis (79.6%, in which OH (25.0%)), direct $O_2$ oxidation (13.7%) and nitrate itself (6.7%). These have been added to the revised manuscript. However, as mentioned in the response to specific comment 3, we are sorry for that it is difficult for us to conduct a kinetic analysis to identify the contribution of $NO_2$ and $NO_2^-$ due to the lack of more parameters.

6. Line 200 - 211: Addition of $(NH_4)_2SO_4$ or $NH_4HSO_4$ can adjust the pH but it can also affect other properties such as ionic strength. How would the authors confirm that other factors are not important? Gen et al. (2019) also investigated the effect of pH on $SO_2$ uptake coefficient. They found that the $SO_2$ uptake coefficient is not sensitive to initial pH and they attributed it to the similar stable final pH even with different initial pH values. Have the authors measured solution pH as the reaction took place? Variation of pH during the reaction may also affect sulfate production rate. Authors should also investigate a slightly higher pH too (e.g., 5 or 6). Such pH falls in the typical pH range during the haze events in China.

(1) We did take other factors into account after the addition of $(NH_4)_2SO_4$ (AS) or $NH_4HSO_4$ (ABS). We calculated the ionic strength of initial AS-adjusted and ABS-adjusted solutions. Through the comparison in Table 1, we can know that there is a positive correlation between ionic strength and sulfate formation rate in AS-adjusted system, while in ABS-adjusted system, the effect of ionic strength seemed insignificant, and the system with high ionic strength does not have high sulfate formation rate. The system (pH=1.80 ABS-adjusted) with the same highest ionic strength as system (pH=3.86 AS-adjusted) have the lowest sulfate formation rate. This result shows that pH is a much more critical factor on the oxidation of bisulfite, that is what we care more about. And more importantly, we attempt to emphasize the key role of the two main components in ambient atmosphere,

(NH$_4$)$_2$SO$_4$ or NH$_4$HSO$_4$. In addition to their own participation in the reaction, they also can regulate pH.

Table 1 Ionic strength of AS-adjusted and ABS-adjusted solutions

| pH | AS/ABS-adjusted | Ionic strength (M) | Sulfate formation rate (µM·min$^{-1}$) |
|---|---|---|---|
| 4.32 | —— | 0.06 | 2.72 |
| 4.03 | AS | 0.0725 | 3.08 |
| 3.96 | AS | 0.135 | 5.79 |
| 3.86 | AS | 0.21 | 6.71 |
| 2.71 | ABS | 0.0725 | 2.17 |
| 2.08 | ABS | 0.135 | 2.62 |
| 1.80 | ABS | 0.21 | 1.13 |

(2) The pH values listed in the Tables are all the initial pH values of each reaction. In addition, we also measured the pH at the end of each reaction. However, no valuable information can be found from them. Moreover, we studied the effect of pH on the sulfate oxidation while Gen et al. focused on the effect of pH on SO$_2$ uptake coefficient. we think that the two are not comparable.

(3) NH$_4$NO$_3$ and NaHSO$_3$ were selected as the two main reactants in our experiment and 4.23 is the maximum pH values of the solution that could be reached under the optimal concentration ratio. Therefore, the experiment with higher pH (5 or 6) could not be performed.

Corresponding revisions have been made in the revised manuscript:

Line 245, the following content has been added into the revised manuscript.

"Furthermore, the addition of (NH$_4$)$_2$SO$_4$ or NH$_4$HSO$_4$ would affect the ionic strength of the solution, and hence the effect of ionic strength on sulfate formation was analyzed based on the ionic strength of initial (NH$_4$)$_2$SO$_4$-adjusted and NH$_4$HSO$_4$-adjusted solutions. As shown in Text S4 (Table S2), it is found that there was a positive correlation between ionic strength and sulfate formation rate in (NH$_4$)$_2$SO$_4$-adjusted system, but in NH$_4$HSO$_4$-adjusted system, the effect of ionic strength seemed insignificant, and the system with high ionic strength did not have high sulfate formation rate. Therefore, this result shows that pH is a more important factor affecting the oxidation of bisulfite when compared with ionic strength."

7. Line 246 – 249: Please refer to Figure 4a and 4b in Gen et al. (2019), their results also suggested that sulfate formation rate during NH$_4$NO$_3$ photolysis is slightly higher than that

during $NaNO_3$ photolysis. The type of cation has little influence on the quantum yield of nitrate photolysis, but it does not necessarily mean that the sulfate yield is comparable regardless of type of cations. As authors suggested in the main text, $NH_4^+$ may play a role in the sulfate formation.

We thank the reviewer for the useful comments and consideration.

In fact, besides the role of $NH_4^+$ in the sulfate formation, we also investigated the effect of the type of cation on the formation of $H_2O_2$. Previous studies haven't detected $H_2O_2$ product during steady-state irradiation of $NO_2^-$ and $NO_3^-$ solutions at $\lambda > 200$ nm, but Wagner et al. found $H_2O_2$ formation in flash photolysis of nitrate ions in aqueous solution (Wagner, et al., Zeitschrift für Physikalische Chemie 1980, 123, 1-33). Mack et al. thought that the combination reaction between two •OH radicals produced by nitrate photolysis is highly unlikely due to the very low concentration and short lifetime of •OH, and then they attributed the formation of $H_2O_2$ observed by Wagner et al. to the $H_2O$ photolysis at $\lambda > 180$ nm (Mack, et al., J. Photochem. Photobio. A. 1999, 128, 1–13). But it is worth noting that most of the nitrates used in their studies are alkali and alkaline earth metal nitrates, such as sodium nitrate solution, potassium nitrate, calcium nitrate solutions, and magnesium nitrate solution. Besides these studies, Yabushita et al. investigated the photolysis of nitrate produced from nitric acid adsorption onto ice surface, and they found that gaseous •OH production should be attributed to the secondary photolysis of $H_2O_2$ produced on ice surface, implying the formation of $H_2O_2$ (Yabushita, et al., J. Phys. Chem. A 2008, 112, 9763–9766). However, up to now the photolysis of ammonium nitrate has still received little attention.

Recently, we found that $H_2O_2$ is difficult to be detected in the photolysis process of sodium nitrate solution, but it is easy to be observed in the solution containing ammonium nitrate, which means that the photolysis behavior of ammonium nitrate solution is different from that of sodium nitrate solution, and thus implies that they would have different effects on the oxidation of bisulfite under light. The effect of cation type on bisulfite oxidation deserves attention. We are preparing another paper on this new finding.

8. Line 263 – 265: Low pH is favorable to the formation of HONO, but the presence of $NH_3$ will increase pH. Any suggestion why the presence of $NH_3$ can promote the hydrolysis of $NO_2$ and induce the explosive growth of HONO?

The $NH_3$ that was discussed here came from the hydrolysis of $NH_4^+$ and was not directly added. As is known to all, the hydrolysis of $NH_4^+$ in the aqueous solution can produce $H^+$ and $NH_3 \bullet H_2O$. The former enhances the acidity of the solution, while the latter produces $NH_3$.

The possible reasons why the presence of $NH_3$ can promote the hydrolysis of $NO_2$ and induce the explosive growth of HONO have been illustrated in the manuscript (line 257-267). The main reasons are: (1) $NH_3$ can promote the shift of chemical equilibrium of the hydrolysis of $NO_2$ (reaction R6) and then enhance the formation of HONO; (2) $NH_3$ can promote the hydrolysis of $NO_2$ and induce the explosive growth of HONO via reaction R6 by reducing the free energy barrier of the reaction and stabilizing the product state (Li et al., 2018a; Xu et al., 2019).

Appropriate revisions have been made in the revised manuscript.

9. Line 287 – 289: Have the authors conducted experiments in the absence of $NH_4NO_3$, i.e., compared Dark+$O_2$ with 2-propanol+ Dark+$O_2$, which is a direct way to investigate the role of $O_2$ in sulfate formation.

Thank you for your suggestions.

We haven't conducted experiments in the absence of $NH_4NO_3$. This is because that what we attempted to discussed here is that the key role of $O_2$ for the process of nitrate photolysis that affecting the conversion of $HSO_3^-$, rather than the direct role of $O_2$ in sulfate formation. Meanwhile, as can be seen from Fig. 1, the direct oxidation of $HSO_3^-$ by $O_2$ was also verified according to S'3 (8.74 $\mu M \cdot min^{-1}$).

10. Line 291 – 293: It is not clear what the authors would like to prove. The role of $O_2$ in the direct oxidation of $HSO_3^-$ by nitrate? Or just want to show the direct oxidation of $HSO_3^-$ by $O_2$? Please clarify the role of $O_2$ in the presence and absence of irradiation.

Here we attempted to discuss the role of $O_2$ in the oxidation of $HSO_3^-$ by nitrate photolysis rather than its direct oxidation to $HSO_3^-$. We pointed out that $O_2$ oxidation is not the main direct contributor on bisulfite oxidation when nitrate photolysis under 313nm UV irradiation, but it could promote the photodegradation of nitrate or react with $O(^3P)$ to form $O_3$ to oxidize bisulfite. However, under dark conditions, $O_2$ directly oxidizes sodium bisulfite and this pathway is the most important way for sulfate formation.

Appropriate revisions have been made in the revised manuscript:

"The inconspicuous inhibition showed that $O_2$ oxidation pathway is not the main direct contributor to the aqueous phase oxidation of bisulfite when nitrate photolysis occurs under 313 nm UV light."

11. Line 297 - 303: How is $O_2$ related to the OH formation from nitrate photolysis? Gen et al. (2019) did not present that OH generation from nitrate photolysis requires $O_2$. The statement presented in Gen et al. is that oxidation of dissolved $SO_2$ by OH radicals requires

$O_2$. Again, it would be useful to compare the contribution of $NO_2$ pathway, $NO_2^-$ pathway and OH pathway to sulfate formation? Gen et al. (2019) found that direct oxidation of dissolved $SO_2$ by $NO_2^-$ is an efficient pathway for sulfate formation.

We have modified the misunderstood sentences. And little relationship between $O_2$ and OH formation from nitrate photolysis was found, which we have already expressed in the manuscript. To avoid ambiguity, we revised the explanations for the key role of $O_2$ on sulfate formation (apart from its direct oxidation). That is, in the presence of $O_2$, $O(^3P)$ will react with $O_2$ to produce $O_3$, thus forming a new oxidation pathway. And we calculated that the OH pathway contributes 25.0% to the sulfate formation under 313 nm UV light. We are sorry for that it is difficult for us to conduct a kinetic analysis to identify the contribution of $NO_2$ and $NO_2^-$ due to the lack of more parameters.

12. Line 309 – 314: In the presence of $O_2$, $O(^3P)$ will react with $O_2$ to produce $O_3$. As displayed in the Figure 1a, a larger difference between S1 ($NH_4NO_3$+Light+Air) and S4 ($NH_4NO_3$+Light+$N_2$) may be also attributed to the $O_3$ pathway.

Thank you for your comments. We added the explanation of ozone pathway to the difference between S1 ($NH_4NO_3$+Light+Air) and S4 ($NH_4NO_3$+Light+$N_2$) under simulated solar irradiation and the difference between S'1 and S'4 under UV light. And, it was also an appropriate explanation for the key role of $O_2$ in sulfate formation (its indirect role rather than the direct oxidation of bisulfite).

13. Line 329: Figure S2 is the "First-order photodegradation of 2NB under 313 nm UV irradiation". The trend of $H_2O_2$ concentration should be in Figure 6. Please revise.

Thank you for pointing this out. It has been revised in the text.

14. Line 331 – 334: It should be noted that reaction of $O_3$ with OH can produce $HO_2$. Hence, there is no strong evidence to conclude that $H_2O_2$ is owing to the recombination of OH.

Firstly, we are sorry for that we have not found the literatures on the reaction of $O_3$ with OH to produce $HO_2$. The only literature we found is that $O_3$ reacts with $OH^-$ to form $HO_2$ under alkaline conditions (Gligorovski et al., Chem Rev. 115, 13051-13092, 2015). This reaction does not occur in our system (pH < 4.32). Secondly, Warneck and Wurzinger reported that the quantum yields at 305 nm for reaction of OH with OH to form $H_2O_2$ in aqueous solution was $9.2 \times 10^{-3}$ $M^{-1}s^{-1}$ (Warneck, P.; Wurzinger, C. J. Phys. Chem. 1988, 92, 6278), while the aqueous-phase oxidation of bisulfite by $O_3$ is rapid, the rate constant of the oxidation of bisulfite by $O_3$ is $3.7 \times 10^5$ $M^{-1}s^{-1}$ (Seinfeld, J. H. and Pandis, S. N.: Atmospheric Chemistry and Physics: From Air Pollution to Climate Change, 2nd Edn.,

John Wiley & Sons Inc., New York, 2006.). Obviously, if $O_3$ is quickly consumed by bisulfite, it will not react with OH. Thirdly, it is generally believed that $HO_2$ mainly comes from the oxidation of CO or VOCs by OH radicals or $O_3$, as well as photolysis of formaldehyde (Lee et al., Atmospheric Environment, 2000, 34, 3475-3494; Schuttlefield et al., J. Am. Chem. Soc., 2008, 130, 12210-12211.). Finally, in PNAS, Lee et al. recently reported experimental evidence that $H_2O_2$ is spontaneously produced from pure water by atomizing bulk water into microdroplets, which does not occur in bulk aqueous solutions (Lee et al., PNAS, 2019, 116, 19294-19298), and they also proposed the formation of $H_2O_2$ from the recombination of hydroxyl radicals generated from the interaction of electric field with $OH^-$. Therefore, the formation mechanism of $H_2O_2$ from the recombination of OH can be supported by some previous studies.

15. Line 352 – 353: Figure S5 shows that the sulfate production rate under 100 mW/cm$^2$ is comparable to that in 50 mW/cm$^2$. Any suggestions?

The experiments with different light intensity were carried out under xenon lamp (simulated sunlight, containing a small fraction (<5%) of UV light), and 30 mM $NH_4NO_3$ solution was used. The experiments only changed the light intensity, other reaction conditions were the same. Considering that bisulfite is oxidized rapidly by the products of nitrate photolysis, such as OH and $H_2O_2$, while nitrate photolysis is slow (the reaction rate constant of nitrate photolysis even at 305 nm UV light is $10^{-7}$ or $10^{-8}$ order of magnitude, see Table 2), this phenomenon reflects that the photolysis of nitrate is a rate-determining step in the whole reaction. That is, when the light intensity is higher than 50 mW/cm$^2$, the rate of sulfate formation is nearly limited by the rate of nitrate photolysis.

16. Line 355: Zhang et al. (2020) found that sulfate production rate will be enhanced during nitrate photolysis in the presence of halide ions. And they suggested that presence of halide ions can enhance nitrate photolysis, as a result, more oxidants will be produced from nitrate photolysis, which will promote sulfate formation further. While the halide related reactions are possibilities, have the authors conducted any simulation to investigate if enhanced sulfate production is attributed to the enhanced nitrate photolysis? In Zhang et al. (2020) halogen chemistry was included in their box modeling and was not found to play an important role in the enhanced sulfate production.

This is a good suggestion.

From the point of view of chemical equilibrium shift, the introduction of halogen ions consumes the oxidizing species produced from nitrate photolysis, which will promote the photolysis of nitrate and produce more new oxidizing species. These new species will promote sulfate formation further. In our paper, we give a qualitative explanation of the

results, but not a quantitative explanation through modelling. Although the modelling may give some valuable quantitative results, we can't give the calculation results in a short time because we lack experience in model calculation. Therefore, we are sorry that we have not been able to explain it by using the results of the model calculation, and your understanding is very much appreciated.

**Review 2:**

The study by Chen et al. explored the influence of nitrate photolysis on sulfate formation from aqueous-phase bisulfite oxidation based on chamber experiments. The role of pH, halogen chemistry and $O_2$ are investigated. This is an interesting topic, and the experiment looks carefully conducted and calibrated. However, the results are poorly organized and reported, with ambiguous data interpretations and conclusions. I don't think the current manuscript meet the standard of ACP, unless it was largely rewritten. Moreover, the authors should treat publications more seriously. Typos are common, but it was very rare to be present at the first line of the abstract—where I believe the "exit" should be "exist". I wonder whether the authors have carefully read the manuscript even once. If even the authors don't want to read the manuscript, neither do the readers.

We are sorry to make so many mistakes. According to your suggestions, we rearranged the structure of the manuscript, carefully proofread the experimental data and their explanations, and provided as many quantitative descriptions as possible to highlight our novelty compared with existing studies. We have seriously corrected the grammatical errors and checked the full text as well. Thanks for your care and patience. The explanations and modifications for specific comments are as follows.

Major concerns:
1. The study titled "Aqueous phase oxidation of bisulfite influenced by nitrate photolysis", and one would expect the study to focus on reactions under light. In this context, the dark condition is expected to serve as a background. However, this study spends quite a lot of efforts emphasizing the importance of nitrate and its synergism with halogen chemistry in sulfate formations even under dark condition. Moreover, the dark condition is apparently not set as the background scenario, as few comparisons are conducted between dark and with light conditions when the other factors are the same. Either the title or the organization and analysis of this study should be modified.

    The title and content of the manuscript have been adjusted as a whole, and more comparisons have been made between dark and with light conditions.

    The title is changed to "Aqueous phase oxidation of bisulfite influenced by nitrate and its photolysis".

2. Line 47-49 and 233-239: The review of aerosol pH is biased. The same research group of Guo et al. (2017) have reported much higher pH levels (even higher than 7) in Beijing in some later studies that they participated (e.g., Shi et al., 2017). In addition, some recent studies have further revealed the driving factors of aerosol pH (e.g., Pye et al., 2020; Zheng et al., 2020), which can explain the difference in reported pH levels. These advances should be included.

   According to your suggestions, the literatures mentioned above have been cited, and appropriate revisions have been made in the revised manuscript.

3. L147-150: The statements are confusing. Which one is insignificant?

   This refers to the experiments under xenon lamp light source, which has an insignificant sulfate yield. To avoid misunderstanding, the sentence "Considering the insignificant increase of sulfate yield, …" is changed to "Considering the insignificant increase of sulfate yield under Xe lamp light source, …".

4. Line 167-169: What's the point / conclusion of the whole part? Is $NO_2$ an important oxidant or not, based on your experiment results?

   Gen et al. reported that $NO_2$ is one of the contributors to sulfate formation. Sarwar et al., (Sarwar et al., Atmos. Environ., 2013, 68, 186-197) reported that the aqueous-phase oxidation of S(IV) by $NO_2$ increases mean winter sulfate by 4-20%. Here we wanted to verify this $NO_2$ pathway for sulfate formation in our reaction system through the observation of the formation of $NO_2$ during nitrate photolysis.

   However, we are sorry for that it is difficult for us to conduct a kinetic analysis or modelling to identify the contribution of the formed $NO_2$ in our aqueous-phase system due to the lack of more experimental data.

   Appropriate revisions have been made in the revised manuscript:

   Line 169: "The formation of gaseous $NO_2$ during the photolysis of nitrate may reveal the aqueous-phase oxidation of bisulfite by $NO_2$ pathway." has been added.

5. Line 176-178: The whole paragraph is repeating existing explanations of potential pathways. But what's the relationship with this study? Do the results support / disagree with any of the pathways? If not, this part should be simplified into one to two sentences. Discussion of existing studies that is not related to your results should not be part of a research article.

   Thank you for your suggestions. According to your suggestions, we have made appropriate modifications:

Line 173: "as well by previous studies" has been deleted.

Line 174-178: the sentence has been deleted.

In addition, on the basis of the existing potential pathways, we conducted more experiments and found some novel phenomena, such as the promotion of sulfate formation by nitrate itself in the dark, the generation of the $H_2O_2$ by the recombination of OH produced by nitrate photolysis, the formation of $O_3$ from the reaction of $O_2$ and $O(^3P)$. And, the role of OH in our study is much higher than that in previous reports. These results are arranged separately in the original manuscript. In order to avoid misunderstanding, we adjust the structure of the article and describe the differences and novelties respectively.

6. Line 187-188: if "nitrate itself can greatly promote the oxidation of bisulfite in the solution under dark condition" already, how important is the photolysis of nitrate? And what's their relative importance? More quantitative analysis should be conducted, as seems all required experiments are already conducted.

Thank you very much for your suggestions. According to the sulfate yield under 313 nm UV light, the roles played by several pathways are as follows: nitrate itself accounts for about 6.7%, oxygen accounts for about 13.7%, and nitrate photolysis accounts for 79.6%. However, under the irradiation of xenon lamp, the weak nitrate photolysis lead to the low sulfate yield. In this situation, the role of nitrate itself in dark is nearly comparable to that of nitrate photolysis (see Fig. 1a). This quantitative analysis result has been added to the manuscript.

7. Line 213: the pH ranges that are achieved by addition of $(NH_4)_2SO_4$, $NH_4HSO_4$ and their mixtures should be stated here.

Thank you for pointing this out. Accepted and modified.

Line 213: The pH ranges (1.80~4.32) are added in the revised manuscript.

8. Line 9-10, 216-219, and 463-465: The behavior of pH in bulk solutions should not be mixed with that of aerosols. While in solutions, the pH can be sensitive to ammonia sulfate concentrations and the relative fractions of ammonia sulfate / ammonia bisulfate added, it was largely controlled by other factors for aerosols (Pye et al., 2020; Zheng et al., 2020). The results should only be used to infer the dependence of reaction rate on pH, not to be interpreted as the dependence of aerosol pH on ammonia sulfates.

As you mentioned, the atmospheric aerosol pH is affected by many factors, such as RH and the compositions and concentrations of inorganic and organic aerosols, not only by ammonium sulfate/ammonia bisulfate. Atmospheric aerosols include aqueous phase aerosols (e.g. fog and cloud droplets), aerosol particles with water film, and so on. In our

study, the aqueous phase oxidation of bisulfite has been investigated. Considering that secondary inorganic ions are one of the most important components in $PM_{2.5}$, we selected ammonium sulfate/ammonia bisulfate to adjust solution pH. For example, secondary inorganic aerosol (SIA: $SO_4^{2-}$, $NO_3^-$ and $NH_4^+$) accounted for 54.1% of $PM_{2.5}$ and 91.0% of the total water-soluble inorganic ions (TWSI) during haze periods, respectively. While during the non-haze periods, SIA accounted for 43.5% of $PM_{2.5}$ and 90.0% of the TWSI, respectively (Kong et al., Sci. Total. Environ. 2018, 634,1192-1204). Also, hygroscopic SIA easily make particles into droplets under high relative humidity. Of course, the role of ammonium sulfate in adjusting the pH of liquid aerosol particles should not be ignored.

In addition, we didn't claim that the behavior of pH in bulk solutions represents the aerosol. What we discussed is only one of the possible cases of that in aqueous phase aerosols. Moreover, it is not mentioned in the manuscript that the pH of aerosol is dependent on ammonium sulfate. We only proposed the crucial role of ammonium sulfate in regulating pH of solutions in the enhancement of aqueous phase sulfate formation, which is of great significance for understanding the behavior of ammonium sulfate in the atmosphere. Therefore, in the revised manuscript, we emphasize that ammonium sulfate regulates the pH of the solution rather than all atmospheric aerosols.

9. Line 223-224: within which pH range?

The pH range here is within 3.86-4.32, which is adjusted by the addition of different content of $(NH_4)_2SO_4$. Corresponding revision has been made in the revised manuscript:
Line 224: "(within the pH range of 3.86-4.32)" is added.

10. Line 225-227: This possible explanation could be checked with simple calculations. Just scale the formation rates to the pH dependence of bisulfite concentrations.

Thank you for your good suggestion. Bisulfite concentrations under different pH were listed as follows. As shown in the following Table, $HSO_3^-$ concentration is greatly reduced (about 62% of the analytical concentration) when pH < 2.08, and thus the sulfate yield was suppressed (The sulfate yield at pH=2.08 in the table is slightly higher than that at pH=2.72, which is probably due to experimental error).

Table 2 The content of $HSO_3^-$ under different pH

| pH | 1.8 | 2.08 | 2.71 | 3.3 | 3.86 | 3.96 | 4.03 | 4.32 |
|---|---|---|---|---|---|---|---|---|
| $c(H^+)$ | 0.0159 | 0.00791 | 0.002 | 0.0005 | 0.000141 | 0.000112 | 0.000891 | 0.00005 |
| A | 0.000459511 | 0.000165399 | 3.00009E-05 | 6.75086E-06 | 1.85374E-06 | 1.4694E-06 | 1.23777E-05 | 6.53358E-07 |
| $\alpha(HSO_3^-)$-Distribution coefficient | 44.98% | 62.17% | 86.66% | 96.28% | 98.88% | 99.09% | 93.58% | 99.49% |
| $HSO_3^-$ (mM) | 13.49 | 18.65 | 26.00 | 28.89 | 29.66 | 29.73 | 28.07 | 29.85 |
| $SO_4^{2-}$($\mu M \cdot min^{-1}$) sulfate formation rate | 5.63 | 13.12 | 10.83 | 19.13 | 33.53 | 28.94 | 15.41 | 13.58 |

*: The results in the table are calculated according to the sulfate formation rate under xenon lamp

11.   Line 243: What's the new aqueous phase oxidation pathways? And what's the relationship of the sunlight discussions with this section (i.e., the effect of pH on sulfate formation)?

(1) We are sorry about that we used the inappropriate words. What we wanted to express here is that the role of $(NH_4)_2SO_4$ and $NH_4HSO_4$ in regulating the pH, which is significant for nitrate photolysis.

(2) In this section, we discussed the effect of pH on sulfate formation by nitrate photolysis. Because our experiments were carried out under acidic conditions, and the ambient atmospheric aerosol particle is usually acidic under severe haze weather as reported. Therefore, the sunlight became an important factor affecting the reaction. However, according to the literature, the photochemical activity is still maintained at a high level in the severe haze weather. Hence, our experiments and conclusion drew here are reasonable.

Corresponding revisions have been made in the manuscript.

12.   Line 246-270: This part seems to argue that cation profile of nitrate is important due to two reasons: (1) $NH_4NO_3$ are with lower pH than $NaNO_3$ at the same concentrations, and (2) $NH_4^+$ can be oxidized by $NO_2$ to help generate OH. For the first explanation, as stated above, the influence of chemical compositions on solution pH should not be equaled to that on aerosol pH. Therefore, if the authors want to prove this explanation, the formation rates should be compared at the same pH, not the same concentrations. For the second explanation, not all spontaneous reactions can happen—the reaction rates must also be considered. Are there any experiments / references supporting that these reactions can happen at a reasonable rate under ambient conditions?

Thank you for your suggestions.

(1) In this part, we found and proposed the important role of cations in the photolysis of nitrate to promote sulfate formation. Therefore, it is necessary to ensure the same initial concentration of $NH_4^+$ and $Na^+$ cations in the solutions rather than the same pH. In this situation, $NH_4^+$ favored a higher sulfate formation for that the hydrolysis of $NH_4^+$ may maintain a stable and low pH of the solution during the process, which is more conducive to the formation of sulfate as described by R3-R10. What we emphasized here is the role of $NH_4^+$ rather than pH. In addition, we didn't claim that the results of the study in solutions are representative of the aerosol. In fact, atmospheric aerosols include aqueous phase aerosols (e.g. fog and cloud droplets), aerosol particles with water film, and so on. What we discussed is only one of the possible cases of that in liquid aerosols.

(2) We agree with you. That is, not all spontaneous reactions can happen—the reaction rates must also be considered.

There are some reports about the thermal reaction of $NH_3$ with $NO_2$ (not the selective

catalytic reduction of nitrogen oxides (NOx) by $NH_3$ ($NH_3$-SCR)), which is kinetically relevant to the deNOx process. The reported reactions are usually carried out at high temperature. In addition, although on the internet there is an experiment about the rapid generation of white smoke by mixing gaseous $NO_2$ and $NH_3$ at room temperature, we found that there are not any formal reports supporting that these reactions can happen at a reasonable rate under ambient conditions. Therefore, we have deleted this content in our manuscript.

Corresponding revisions have been made in the manuscript and supporting information.

13. Line 273-284: What's the point of the long discussions of this part? Proving that 2-propanol can serve as *OH scavenger?

Our purpose is to discuss the role of oxygen and OH in the formation of sulfate. 2-propanol is only used as a tool for its inhibition on oxygen and OH. To avoid misunderstanding, this section is integrated into Section "*Aqueous Oxidation of Bisulfite by Nitrate Photolysis*".

14. Line 294-297: If "$O_2$ has little effect on the generation of OH by nitrate photolysis", how is it important in sulfate formation with nitrate photolysis? Any explanation of the potential pathways?

$O_2$ has little effect on the generation of OH by nitrate photolysis, but has an important effect on the formation of sulfate. It is reasonably inferred that $O(^3P)$, one of the products of nitrate photolysis ($NO_3^- + h\nu \rightarrow NO_2^- + O\,(^3P)$), will react with $O_2$ to produce $O_3$ ($O\,(^3P) + O_2 \rightarrow O_3$), thus becoming an important contributor for sulfate formation.

15. Line 317-326: Rewrite this paragraph to make the points clearer.

We rewrite this paragraph as follow.

"Atmospheric $H_2O_2$ plays an important role in sulfate formation, and its formation has attracted much attention. Previous studies haven't detected $H_2O_2$ formation during steady-state irradiation of $NO_2^-$ and $NO_3^-$ solutions at $\lambda > 200$ nm (Daniels et al., 1968; Shuali et al., 1969; Mark et al., 1996), but Wagner et al. once found $H_2O_2$ formation in flash photolysis of nitrate ions in acidic aqueous solution (Wagner et al., 1980). Yabushita et al. once again found $H_2O_2$ formation during the photolysis of nitrate originated from nitric acid adsorption under low-temperature ice conditions, and they attributed its formation to the recombination of the •OH produced by nitrate photolysis (Yabushita et al., 2008). However, Mack et al. thought that the combination reaction between two •OH produced by nitrate photolysis is highly unlikely due to the very low concentration and short lifetime of •OH, and they attributed the formation of $H_2O_2$ observed by Wagner et al. to the $H_2O$ photolysis at $\lambda > 180$ nm (Mack and Bolton, 1999). The formation of $H_2O_2$ during the photolysis of nitrate remains controversial, which is worth exploring further."

16.  Line 336-342: Is the higher $H_2O_2$ under lower pH totally due to the higher OH productions under lower pH, or is it due to that $H_2O_2$ formation is also favored under low pH even assuming same OH concentrations?

As is shown in Table 1, the quantum yield of OH obviously enhanced as pH decreased, illustrating that the lower pH favored the higher OH production. Meanwhile, higher $H_2O_2$ formation was also occurred. We can't exclude the role of lower pH on higher $H_2O_2$ formation. Roth et al. once studied the $H_2O_2$ formation by the combination of OH in the radiolysis of water. They verified that higher primary $H_2O_2$ yields were occurred at a low pH (<4), while the yields decrease at higher pH too.

In addition, it is worth pointing out that $H_2O_2$ increased first and then decreased during UV irradiation, which may indicate that the secondary photolysis reaction occurs during $H_2O_2$ formation (see Fig. 6).

17.  Line 351: how high is the "higher" light intensity? And how sensitive is the formation rate on light intensities?

The experiments with different light intensity were carried out under xenon lamp (simulated sunlight, containing a small fraction (<5%) of UV light), and 30 mM $NH_4NO_3$ solution was used. The experiments only changed the light intensity, other reaction conditions were the same. Fig. S5 shows the results of the effect of light intensity on aqueous-phase sulfate formation. Firstly, as can be seen from Fig. S5, the sulfate production rate under 100 $mW/cm^2$ is comparable to that in 50 $mW/cm^2$. Considering that bisulfite is oxidized rapidly by the products of nitrate photolysis, such as OH and $H_2O_2$, while nitrate photolysis is slow (the reaction rate constant of nitrate photolysis even at 305 nm UV light is $10^{-7}$ or $10^{-8}$ order of magnitude, see Table 2), this phenomenon reflects that the photolysis of nitrate is a rate-determining step in the whole reaction. That is, when the light intensity is higher than 50 $mW/cm^2$, the rate of sulfate formation is nearly limited by the rate of nitrate photolysis. Secondly, the sulfate formation rate under 8, 50 and 100 $mW/cm^2$ was 13.58, 22.56, 22.36 respectively. We found that the increase of sulfate formation rate is obvious when the light intensity increases from 8 $mW/cm^2$ to 50 $mW/cm^2$.

18.  Line 367-391: which steps are proposed in this study? Judging from the manuscript, it seems like all the reactions are already proposed by others, and the study here is just trying to combine them in different ways. If not, clearly state the new pathways/steps proposed in this study and provide the experimental evidences.

As explained before, on the basis of the existing potential pathways, we conduct more experiments and found some novel phenomena. Firstly, in view of the formation pathway of

bisulfite, we found the promotion of nitrate itself on sulfate formation in the dark, and the generation of the $H_2O_2$ (Fig. 6) by the recombination of OH produced by nitrate photolysis. Thirdly, we tried our best to conduct more quantitative descriptions, including the contribution of several reaction pathways to sulfate formation. For example, under 313 nm UV light, the contribution of OH in our study is much higher (25.0%) than that in reported paper (<1%). Moreover, when studying the effect of pH on sulfate formation by adding ammonium sulfate and ammonium bisulfate, we found and proposed the existence of optimal pH (~3.86) value and the important role of $NH_4^+$ in sulfate formation. And, halide-induced enhancement of sulfate formation by nitrate photolysis owing to halide photochemistry and the redox cycle of halogen was proposed for the first time in this study.

19. Line 480-485: This implication part seems abrupt and over-interpreted.

This part of implication has been modified appropriately.

Minor concerns:

There's a lot of grammar errors in this study. A thorough langrage editing is suggested. Some examples follow.

1. Line 1, "exit" should be "exist".
2. Line 145, ", see Text S1 in Supporting information, hereafter" is added.
3. Line 145-155: Missing reference of this statement.
4. Line 169: "…stable rate" is changed to "…stable formation rate".
5. Line 181-182: "oxygen" is changed to "$O_2$".
6. Line 201: "then" should be "thus".
7. Line 219, 295, etc.: "on the one hand" should be "on one hand". In addition, after "on one hand" there should always be an "on the other hand", which is not seen in, e.g., line 295.
8. Line 221: "may because" is wrong in grammar.

Thanks for your careful comments. The mistakes listed above have been modified in the manuscript and we checked the whole contents more seriously as well.